# The Influence of Historical Irrigation Canals on Urban Morphology in Valencia, Spain

**Fumiko Ikemoto [1,*], Kosuke Sakura [2] and Adrián Torres Astaburuaga [3]**

[1]   Independent Researcher, 12043 Berlin, Germany
[2]   Department of Architecture, Shinshu University, 3 Chome-1-1 Asahi, Matsumoto, Nagano 390-8621, Japan; kosuke_sakura@shinshu-u.ac.jp
[3]   Ecole Urbaine de Lyon, Université de Lyon, Atrium Building, Domaine Scientifique de La Doua, Boulevard du 11 Novembre 1918, 69100 Villeurbanne, France; a.torres-astaburuaga@universite-lyon.fr
[*]   Correspondence: fumiko.ikemoto64@gmail.com

**Abstract:** As one of the fundamental natural resources of life, water and its management within ecosystems has always been the most crucial aspect of any settlement. Prior to urban modernization, water was sourced upstream from rivers or groundwater, supplying settlements, with the runoff being drained further downstream or to sea, creating a series of water flows; our livelihood coexisted with this series. In the rapid city growth led by modernization, due to the creation of uniform and homogeneous new urban areas, water flow became separated for each purpose and began to be specifically manipulated for, and by, human society. This study was designed as one of a series of research projects aiming to highlight the relationship between the historical hydraulic systems and the more recent urban spatial structure, with the focus on Valencia, one of the medium sized cities in Spain. Valencia is ideal as a case study due to the historical mechanisms of hydraulic systems still partially in use, such as irrigation canals in its agricultural regions and sewage canals in its urban areas. In more recent years, the ancient canals and the rivers that were neglected or buried, due to pollution and/or flooding concerns, began to regain significance in the face of the growing interest in and necessity of restructuring green spaces in the city as well as the preservation of the city's unique identity and history, along with its remaining/evolving ecosystems. The purpose of our research is to interpret the interaction between Valencia's urban morphology and its historical irrigation systems, particularly its waterways. The target period is from the modernization in the 20th century to their present conditions.

**Keywords:** hydraulic system; irrigation canals; urbanization; Valencia



## 1. Introduction

### 1.1. Background

Valencia is one of the mid-scaled cities located on the west coast of the Mediterranean Sea, tracing the roots of its agricultural activity back to the time of the Roman Empire [1]. Structured and developed in the Arab period (from the 9th to 13th centuries), it flourished throughout the Late Middle Ages, as the city acquired relevance as a silk producer in the 15th and 16th centuries. The landscape created by the agricultural activity reflects the intensity of the occupation of the territory and its economic activity. This landscape, shaped by rural activities that depended on complex vernacular hydraulics systems, influenced the morphogenesis of proto-urban aggregations that became the city of Valencia, while maintaining a constant relationship with the countryside [2].

Today, the irrigation canals (acequias) in these hydraulic systems, diverted by means of weirs (azudes) from the flow of the Turia river, cross the city in a non-visible way in order to irrigate peri-urban agronomic, fertile and productive areas, reaching to the coastline [3]. The contribution of water to the soil through flood irrigation, and the discharge of water to the wetlands, such as the Albufera in the south, plays a singular role for groundwater

recharge, the maintenance of the hydric phreatic water reserve [4], the salinity balance of coastal soils, and therefore its fertility.

This irrigation systems were originally introduced in the city by Roman colonists, who founded a city called Valentina in 138 BC. In the Islamic period between the 9th century and the 13th century [3], the water network of the city was reorganized and extended to the surrounding agricultural areas, together with city expansion. These irrigation systems introduced in the Islamic period were based on the technology shared by different regions in the Iberian Peninsula, southern Europe, Northern Africa, and Middle East, as a result of the Muslim expansion in these regions at that time [5]. During this period, there was a transfer of knowledge, with trips back and forth between cities in the Maghreb, such as Marrakech, Fez or Meknes [6], and the knowledge that was being developed by hydraulic engineers in cities such as Valencia or Granada.

However, as the city grew and expanded after industrialization, rural areas were urbanized, and some of the canals were covered, especially those that crossed urban areas. Today, most of the canals are not readily visible in urban contexts; nevertheless, they have been subtly inherited and incorporated in urban morphology, especially inside of the historic center, such as in the layout of roads, urban alignment, block interiors and plot border, or in the form of urban green spaces and its irrigation system [5].

In Valencia, in spite of the fact that the city separated this natural-based historical technology from the modern city and disregarded its value, the hydraulic systems have somehow been preserved and contributed to create the unique landscape of Valencia [7,8]. Today, this indirect preservation of historical elements can be understood as an opportunity to rethink the relationship between the city and water, the history and the community.

Furthermore, the Tribunal de les Aigües de València (Valencian Water Court), the oldest national legal institution still in existence, responsible for settling disputes over the distribution and use of irrigation water, was chosen as an intangible cultural heritage by UNESCO in 2009.

Additionally, the canals and their related constructions have been nationally protected since 2014, both for their essential water supply function for agricultural production, related to the socioeconomic development of the region, and for their role as true articulators of the territory understood as cultural landscape [9].

The historic hydraulic systems, being socio-ecological resources, were crucial in the formation of the cultural landscape of Valencia, shaping the ecological and artificial process of the city's development, transforming the natural resources into an expression of culture. Interpreting the urbanization through the preservation of these historic irrigation canals provides significant suggestions for contributing to the debate on urban theory and the practice of it toward a sustainable urban development.

Several studies describe the history of irrigation canals in Valencia. Javier Martí, in his study "Las Venas de la Metrópoli. Séquies, Rolls i Cadiretes en la Ciudad de Valencia" (2007) [3] examines the relationship between the historical canals and the medieval urban structure. In turn, the study by Carlos Sanchis Ibor, "Acequias, Saneamiento y Trazados Urbanos en Valencia" (2002) [10] illustrates the reduction of vernacular hydraulic systems in pursuit of a modernization of the infrastructure in relation to urban expansion.

Regarding the influence of the irrigation systems on the present urban structure, in the paper by the co-author of the present study, "CIUTAT VELLA VALÈNCIA. Memoria del agua, estratigrafía urbana, reactivación de uso." (2018) [7], Adrián Torres Astaburuaga demonstrates the relationship between the urban morphogenesis of Valencia and the hydraulic system in the city center. Using geological and landscape analysis, this paper shows how the pre-anthropic geography and the vernacular hydraulic system influenced the formation and urban development of the city center by the aggregation of rural and urban landscapes.

Moreover, the paper by Kosuke Sakura "The Relationship between the Transformation of Urban Morphology by the Modernization and the Irrigation Canals. Focused on Spanish Middle Size City Valencia" (2015) [8] analyzes the case of the first urban expansion of

Valencia after the demolition of the old city wall in the late 19th century. This paper assumes, from the perspective of geological and urban history, that the scale and pace of urban expansion meant that irrigation canal lines were maintained in the urban areas of the city.

More recently, and with a similar methodology, the study by Rafael Temes Cordovez, "Caminos, acequias y parcelas en la forma urbana de la ciudad de Valencia" (2020) [11] focuses on the transcription of the vernacular agricultural plots and superimposes them on the current urban structure.

In the present research, we focus on the transformation of Valencia's irrigation canals during modern urbanization in the 20th century to discover the relationship between them and the urban spatial structure developed in this period. Then, we analyze how these hydraulic systems have influenced the formation of the current urban landscape through modernization through which the city experienced change from being an organically based society to an industrial society based on hydrocarbons.

### 1.2. Retrospective Hypothesis for a Retrospective Urban Redevelopment Model of Valencia

The existence of these remnant historic canals is known throughout southern Iberia and Italy, North Africa, the Middle East and Cyprus; they are also found in Central Asia, western China and in dry regions of Latin America [12]. Especially regarding the irrigation canals developed together with Muslim expansion, studies on Granada (Spain), Sicily (Italy) and Marrakech (Morocco) stand out. However, the number of studies regarding the influence of the historic canals and present urban structure is limited. They focus on irrigation canals from the viewpoint of geology and archaeology, and highlight their influence on the soil systems, historic rural landscape, agricultural diversity or historical management system of the water [13–16]. The study by David Arredondo Garrido, "Espacios en tránsito: transición agro-urbana en la Vega de Granada" [17] describes the urban morphology focusing on the transition of the landscape of La Vega in Granada from rural to urban, yet the direct influence of historic irrigation canals on the present urban structure is not featured.

In addition, some cities, such as Fez, have several studies [18] focused on the influence of historic hydraulic systems on urban morphology, but mainly the studies have a historiographic point of view regarding the potentialities between the hydraulic systems and the current urban structure. The case of Lyon is extremely interesting and has been extensively studied [19] as a case of a river city that has to manage violent floods, and therefore, defines and shapes itself with water as both an ally and enemy at the same time [20]. The influence of geomorphology on the city of Valencia, from an archaeological point of view, has also been extensively studied [21]; there have also been a large number of studies relating to the restitution and analysis of irrigation canals in the city of Valencia [22] and in many others [23].

All these research methods are comparable to the present study; however, the former research is based on the macro perspective analysis gained by the document survey. There are fewer studies combined with the micro perspective analysis based on the field survey.

Our research presents a methodology that aims to combine the geomorphological conception intrinsically implicit in the anthropic but sensitive to topography irrigation canals, with a reading from the current state, with an orientation of recovery and implementation of actions oriented to the operation also in the urban environment, associating them by means of a continuity solution with the system of green spaces. Our approach is based on archaeology and historiographical reconstruction but is superimposed on the present stage to assess the extent to which these hydraulic elements can articulate a fertile network of open spaces. Tradition, territory, biotopes and society are allied with contemporary urbanity to give meaning and coherence to contemporary urban contexts.

Furthermore, together with the above studies conducted by the co-authors, we wish to enunciate through the continuous construction of the "Retroprospective Valencia Model" a comprehensive understanding of the peculiar relationship between the historical hydraulic

systems and the urban morphology, and the green space system of Valencia, in order to rethink the priority parameters for planning the city of the future.

This study adds a further piece to the two previous co-author studies written in Spanish and Japanese, a series of research studies that aims to explore the influence of vernacular hydraulic systems on the urban morphology of medium-scale coastal cities in the Mediterranean Arc.

## 2. Materials and Methods

The irrigation canals, as well as the pre-anthropic geomorphology (the logic of surface runoff, the old branches of the river, the ravines, etc.) became the basis of the urban structure, especially in the historic city center. Today, the presence of the historical waterways in urban areas has been minimized, and it is very rare to find open sections of the canal. However, the previous study shows that the presence of historic irrigation canals can be intuited, even in the urbanized area at the periphery of the historic center, as well as the canal existence, which influenced the direction of urbanization before modern development.

Our research examines the transition of the link between urban morphology and the historical irrigation systems, especially during the period of modern urban development. We also examine how these canals were inherited in the present urban structure.

The methods used for this purpose are the document survey and field survey. The document survey was carried out in order to elucidate the influence of the urban development on the historic canals, followed by the field survey in order to interpret the influence of the irrigation canals on the current urban structure.

In this article we focus on slightly suburban areas that were identified as areas for further study, according to our previous research [7,8], within the city of Valencia. The targeted area has potential to continue building a prospective diagnosis that allows a global understanding of the relationship between the urban development of the city and the historic watercourses.

### 2.1. Study Area: Two Functioning Irrigation Canals Crossing Rural, Urban and Peri-Urban Areas

In order to highlight the relationship between the historic irrigation canals and urban development during the period of modernization, we analyzed the historical changes of two historical irrigation canals: the "Acequia de Mestalla" flowing through the south of the city, and the "Acequia de Favara" flowing through the north. For the present study, we chose these two canals among the eight historical canals because they both flow through rural, urban and peri-urban areas, and are therefore relevant for investigating the influence, persistence and potential recovery of these hydraulic systems in urban development. These two canals maintain their function of supplying water to the agricultural areas near the river intakes upstream and the coast downstream, crossing the urban context of the city in between.

The first canal studied, the "Acequia de Mestalla", starts from the old diversion dam, the "Azud de Mestalla", to the northwest of the city, from the right bank of the River Turia [18]. The irrigation channel was historically used to irrigate agronomic areas in the northern part of the city bordering the urban environment to later extend its irrigation perimeter to the coastal area, flowing into the Mediterranean Sea. It should be noted that the ditch preserves sections whose structure dates back to the 13th century [9]. Along its route, we also find historical elements of relevance and of heritage and ethnological value, such as the old mill "Molí d'Alters" and the farmhouse "La Alqueria Fonda". In the modern urban expansion area, to the east of the city, the irrigation stopped when the urban transformation of the city toward the sea began in the 1960s, consolidating the urban development and, therefore, the significant obliteration of these elements studied toward the 1990s.

The other canal of vernacular origin studied in this research is the "Acequia de Favara". It originally started from the disappeared "Azud de Favara", and nowadays its flow is derived from the river by means of the modern dam "Azud del Repartiment". The

hydraulic canal was used to irrigate along the right bank of the Turia in the south-west, with diversions running through the historic city center itself. The canal then runs southwards to the Albufera rice fields [24]. The historical diversion weir has disappeared, due to the transformation of the urban course of the Turia as it passes through the city into a linear garden with no water flow [24]. This canal, one of the oldest, given its proximity to the historic center, is also one of the most damaged, both by modernization itself and by the aforementioned implementation of the Turia Gardens in the old urban riverbed from the 1960s onwards. It was declared an Asset of Cultural Interest on 7 October 2004 [9].

Figure 1, below, shows the irrigation canals according to the catalog published by the city hall [9], "Acequia de Mestalla" and "Acequia de Favara" in a continuous line that flows into the south and north part of the old city wall of Valencia, and which experienced urban development of modernization after the Spanish Civil War, along with six other irrigation canals in a broken line. "Acequia de Rovella" which runs through the city center, was not covered in the present research in order to focus on the peripheral area of the city center that urbanized in the latter half of the 20th century.

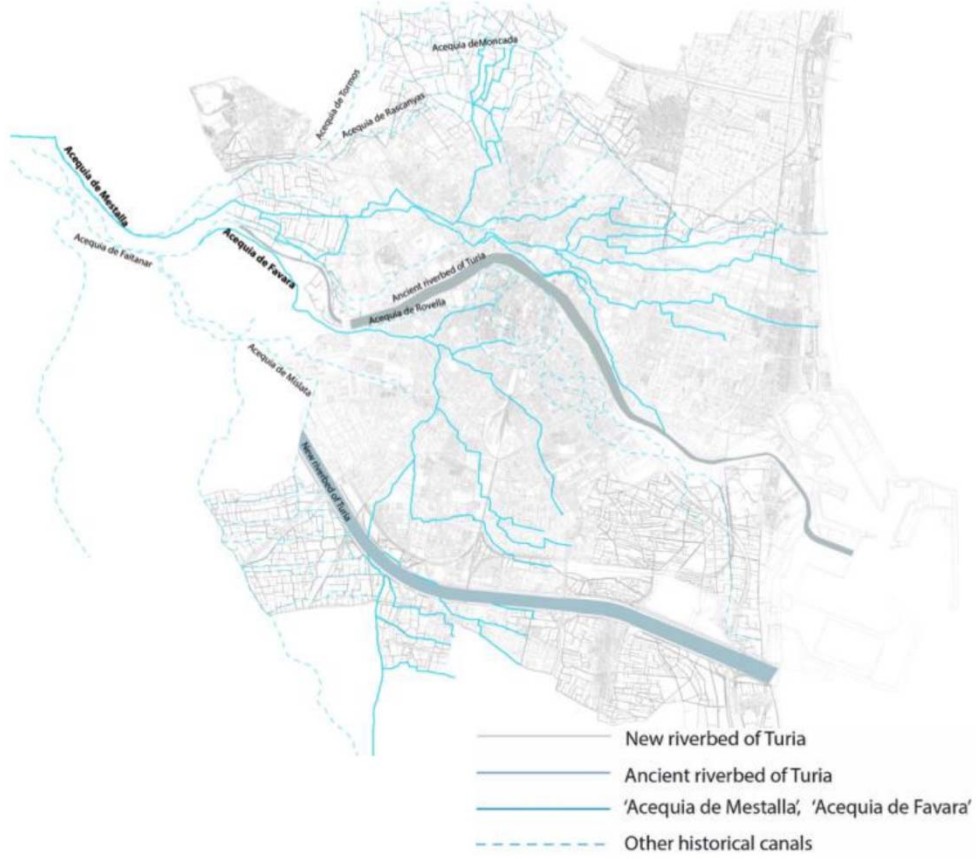

**Figure 1.** Protected irrigation canals and present urban structure.

### 2.2. Target Period or Study: Urban Modernization and Landscape Obliteration

In this paper, we focus on the transition of the relationship between urban structure and the irrigation canals in the 20th century. In Valencia, the speed of urbanization was accelerated in the 1950s because the construction area was expanded largely after the 1950s and the population rapidly increased at the same time [25]. Therefore, it is estimated that there are differences in the degree of the influence of urban development on the irrigation canals and the historic organic matrix of geography and society, before and after 1950.

The city of Valencia was founded by the Romans in 138 BC. From that time onwards, the Romans settled and consolidated themselves as a fortified settlement on a fluvial island provided by the braided course of the River Turia [26].

In the period of the Arab dynasty between the 10th and 11th centuries, the fractal irrigation system was developed and consolidated [27]. These systems are complex interconnected networks that interweave the pre-existing orography to supply water to the territory, as well as to drain the surplus water in flood-prone areas, using only the force of gravity. The Arabs implemented this model based on their own technology, but there was always a reinterpretation and reuse of previous infrastructures, notably those of Roman origin [28].

In the 13th century, after the Christian conquest of the city, the Islamic wall was demolished, and a new wall was built on the previous city area. The hydraulic layout within the enclosure was transferred to the urban road layout [3,10,29,30], and the rural plots were initially transcribed into urban plots [31]. The city expanded based on the local geomorphology and the structure of the irrigation canals. Figure 2, below, shows the expansion of the territory of Valencia until the construction of the Christian wall in the end of the 14th century.

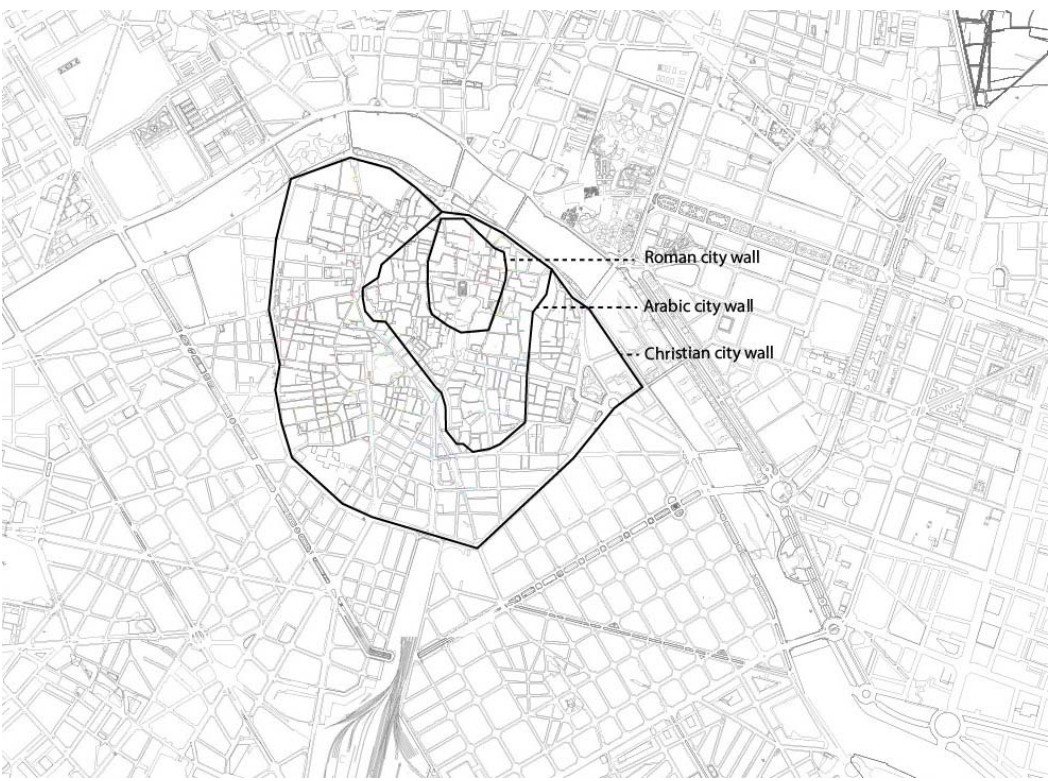

**Figure 2.** The city walls created by the expansion of Valencia from the roman city to the Christian city.

At the end of the 19th century, the demolition of the Christian walls initiated another process of urban expansion. The new areas were created in a grid pattern, partially inheriting the existing town blocks, and the existing irrigation canals were mostly obliterated [32]. Then in the 20th century, the city was developed further, especially after the Spanish civil war (1936–1939). A series of urban planning was approved based on future population and economic growth projections for land-use zoning. Moreover, in 1958, one year after the terrible flood, the city decided to divert the flow of the river Turia to the south of the city, leaving the original river course dry as it passed through the city center, because they considered that the river caused the disaster rather than the impermeabilization and deforestation. As a result, a new urban border in the south was created—a new river channel consisting of a concrete channel flanked by highways. This new south border divided the lines of historic irrigation infrastructure and partially destroyed them. This plan proves that the city paid less attention to the original landscape of Valencia, and

separated the river and the historic irrigation systems, which had established the original landscape, from the modern city.

### 2.3. Document Study: Recomposition of the "Palimpsest" from the Cartographies

In order to interpret the process of urbanization, we superimpose the historical cartographies of the city [33–35] and reconstruct the urban "palimpsest" of Valencia, focusing on the scale of urban expansion and the development of the urban space structure. Then, we also transcribe the irrigation canals' layouts depicted in the cartography in our urban palimpsest. This exercise allows us to understand the extent of urban development and the degree of preservation and obliteration of the irrigation canals in each phase of urban development.

In the present research, the transformation of the canals during the period of urban modernization and expansion in the 20th century is highlighted. Thus, the urban palimpsest in the present study is composed of the cartographies of 1883 to examine the urbanized area and irrigation canals' layouts before the 1900s; the cadastre created between 1929 and 1944 to review the urbanized area and irrigation canals' layouts in the first half of the 20th century; and the cartography of 2009 to analyze the extent of urbanization and the canals' layouts until the 2000s. In this way, the scale of urban development in each period and transformation of the historic irrigation canals are examined. Subsequently the transition of the relationship between urban morphology and irrigation canals before, during and after the modern development of the first half of the 20th century in Valencia is analyzed.

We then refer to the aforementioned official catalogues of Cultural Heritage [9] published by the city council, which contain the map of preserved canals, to define the location of the irrigation canals, which are still in use for irrigation today. We create two urban palimpsests for each of the irrigation channels: "Acequia de Mestalla" and "Acequia de Favara".

### 2.4. Field Study: Identification of the Hidden Waterways

The current state of the irrigation canals within the current urban structure was also investigated from a micro perspective by means of a field study. We retranscribed the historic irrigation canals of "Acequia de Mestalla" and "Acequia de Favara" on the contemporary city plan, from the mother canal and its successive fractal derivations, to understand the current condition of buried channels in today's urban context.

For the development of the micro-scale survey, we focused on the main protected route of the two irrigation systems, which run through the rural area and urban area, then flow into the ocean. Then we highlighted the features in the urban spatial structure shaped by these canals, such as the streets or plot inner partitions that coincide with the canal lines, level differences, bumps, or paths in green areas. Figure 3, below, shows the target areas of the field survey.

The field study complements the historical documentary study. The historical canals are not visible in today's urban structure as waterways. In other words, it is impossible to distinguish the characteristics created by the canals by means of document studies after the canals are buried underground. The objectives of the field study are to focus on the characteristics that are not documented as a lineal or planar watercourse, as well as to interpret the current condition of the target areas, which is constantly changing.

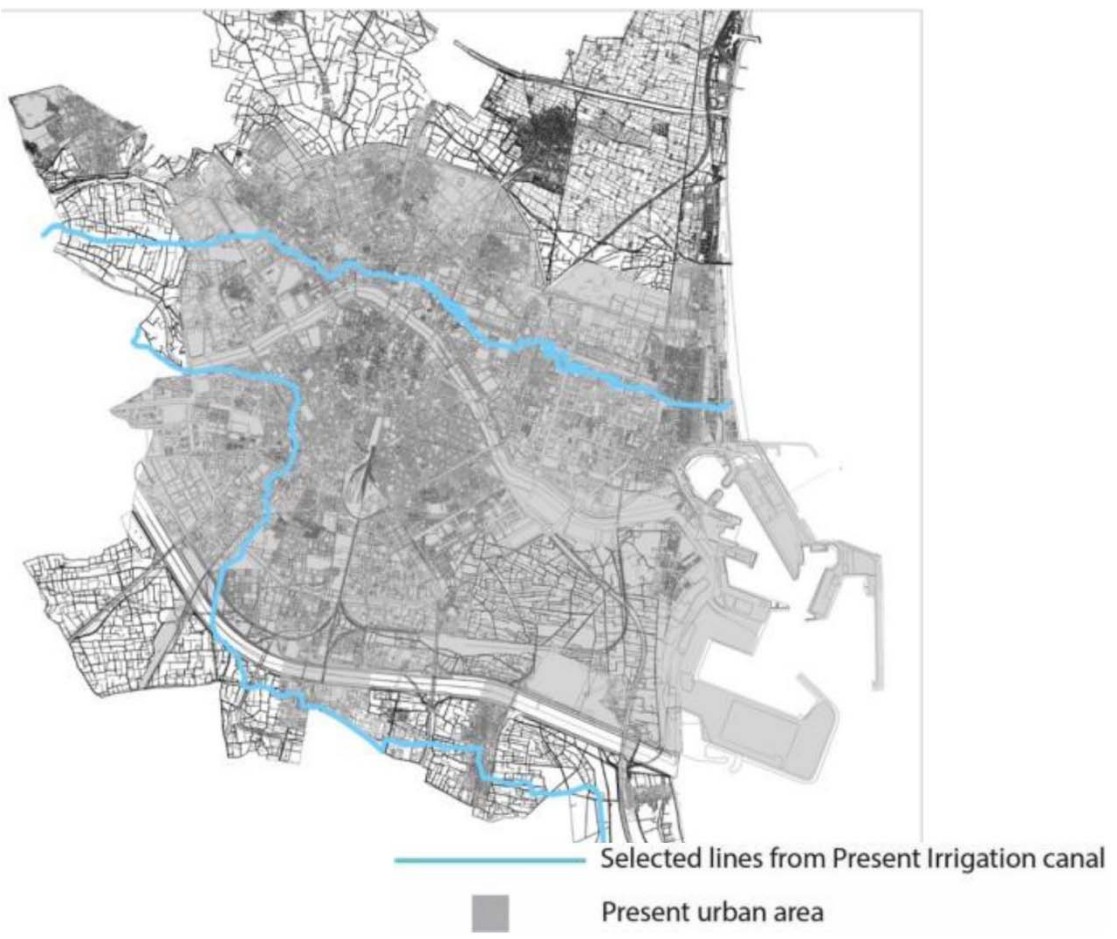

**Figure 3.** The target area for the field survey and urbanized area.

### 3. Results

*3.1. Macro Perspective Analysis: Scanning the Urban Palimpsest*

The macro perspective analysis consists of transcribing the canal traces from the historical maps and comparing their transformation with the evolution of urban spatial structure, thus being able to understand the process of urbanization in relation with the evolution of the hydraulic systems. Figure 4, below, shows the transformation of the layout of the irrigation canals of "Acequia de Mestalla", together with the three phases of urban development until 2010; until 1899; from the 1900s to 1940s; and from the 1950s to 2000s. An adjustment or correction was made by transcribing historical maps, notably the 1883 cartography, as these maps clearly do not have an exact geometrical correction. We then modified the layout of the irrigation channels, sometimes adjusting them to the traces confirmed today by means of institutional cartographic information, as well as interpreting the guidelines of these canals according to the streets, blocks and urban structure up to the point where they flow into the coast.

From this analysis, we discovered that the transformation of the historic channels—obliteration, preservation and expansion—are closely related to the phase of urban development in the area studied.

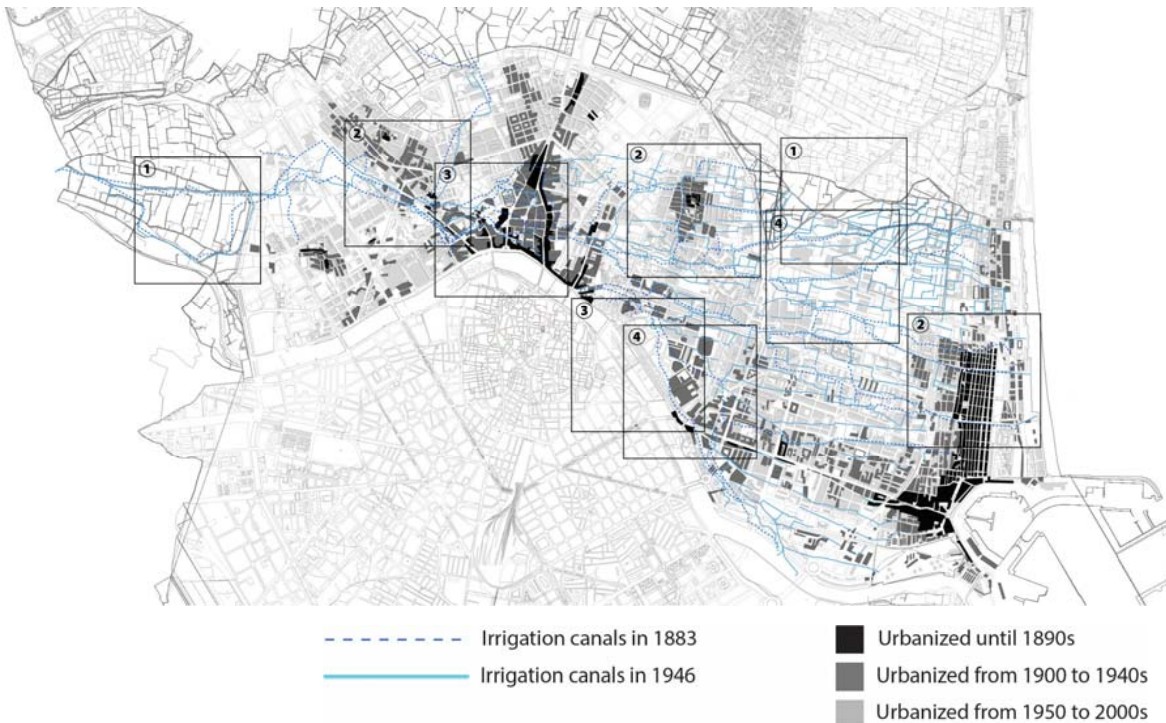

Figure 4. (①–④) The transformation of the irrigation canals and the phases of urbanization acequia de Mestalla.

Regarding the canal "Acequia de Mestalla", in the areas labeled ① in Figure 4, there is a direct preservation of the canals; the shapes of the canals in 1883 are indicative of the shapes of them in 1946 prior to 20th century urbanization. Figure 5 shows the transformation of the irrigation canals in detail in the areas ①. This is found to be due to the preservation of peri-urban agricultural plots that are irrigated through the canals. These areas have a degree of environmental protection as a special ecological and agricultural area in the current planning [36]. Thus, there is significant preservation of both the agricultural parcel and the hydraulic system layouts. These are plots located to the west, upstream, in locations close to the diversion dam, or downstream, in fields close to the coastline where the irrigation channel is uncovered after having crossed urban areas.

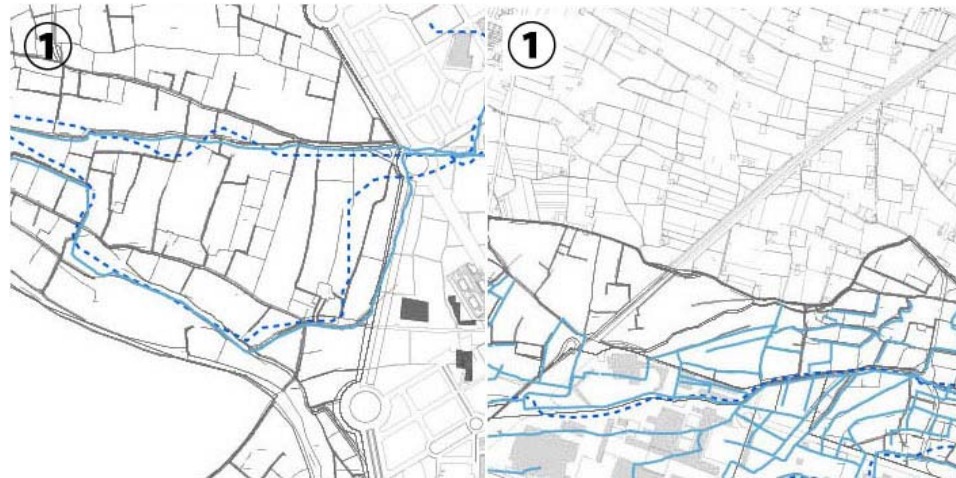

Figure 5. The areas labeled ①, the canals preserved their layout from 1883 to 1946. The canals in these areas are still used for agricultural irrigation systems.

Figure 6 shows the transformation of the irrigation canals in detail in the areas labeled ②. In these areas, the historical canals were not changed in urban structure in the urban development until 1950. In these parts, the new plots were constructed, expanding the existing settlements around the city center. These expansions are based on the extension of the existing road networks and rural crop layout, along which the canals flow. As a result, the urban spaces were developed based on the historic irrigation canals. In addition to this, the irrigation canals used to define the border of the expansion of the settlements in some parts; since then, these canals have remained untouched.

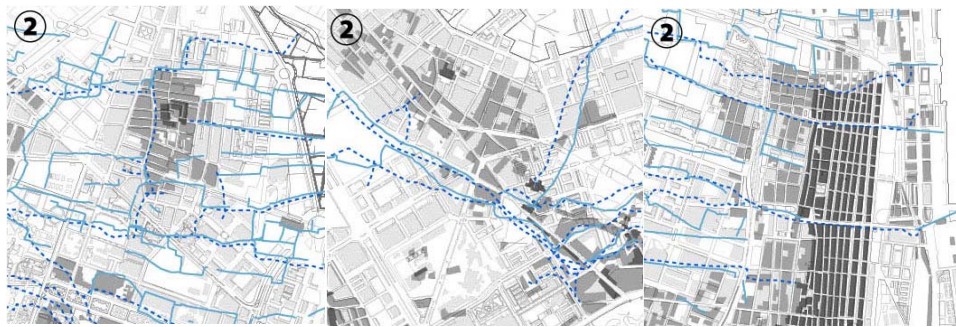

**Figure 6.** The areas labeled ②, the canals preserved in urban space in 1946.

Figure 7 shows the transformation of the irrigation canals in detail in the areas labeled ③. In these areas, the historic canals from 1883 were obliterated by 1946. These areas were urbanized greatly in the first half of the 20th century as public areas or for transportation; therefore, it is estimated that the canals were not considered important elements for the urban space that was in demand for the economic growth of the city.

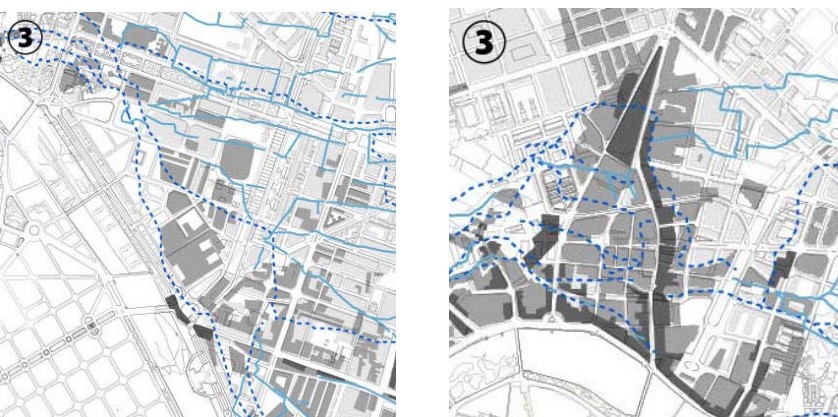

**Figure 7.** In the areas labeled ③, the canals disappeared during the period between 1883 and 1946.

Figure 8 shows the transformation of the irrigation canals in detail in the areas labeled ④. In these areas, the canals have preserved their layout of the 19th century layout and they are even extended. The areas in which the canals were extended are located in the areas between the historic center and the newly built area in the first half of the 20th century. It is estimated that the canal networks were developed at that time for newly built settlements using water networks in suburban cultivation areas. Thus, the development of the canal network and urban configuration were closely linked to each other.

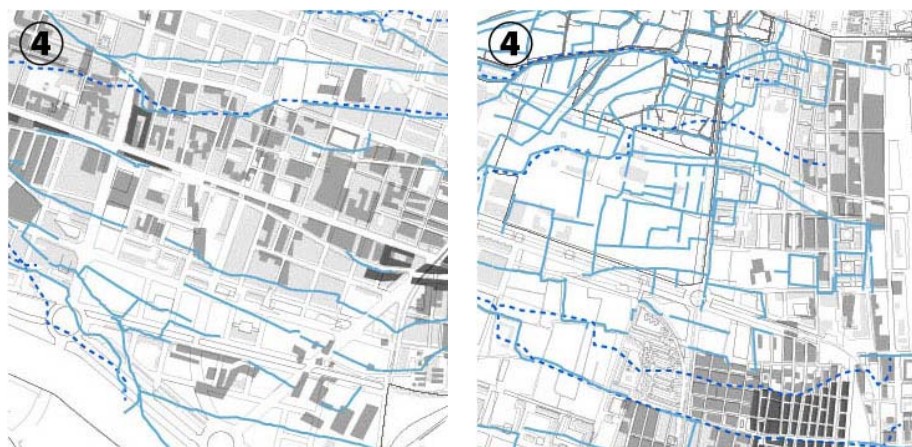

**Figure 8.** In the areas labeled as ④, the canals developed during the period between 1883 and 1946.

Figure 8, below, shows the transformation of the layout of the irrigation canals of "Acequia de Favara", together with the three phases of urban development until 2010; until 1899; from the 1900s to the 1940s; and from the 1950s to the 2000s. Regarding the canal "Acequia de Favara", the areas labeled as ⑤ in Figure 9, the 1883 canal layouts have been largely preserved.

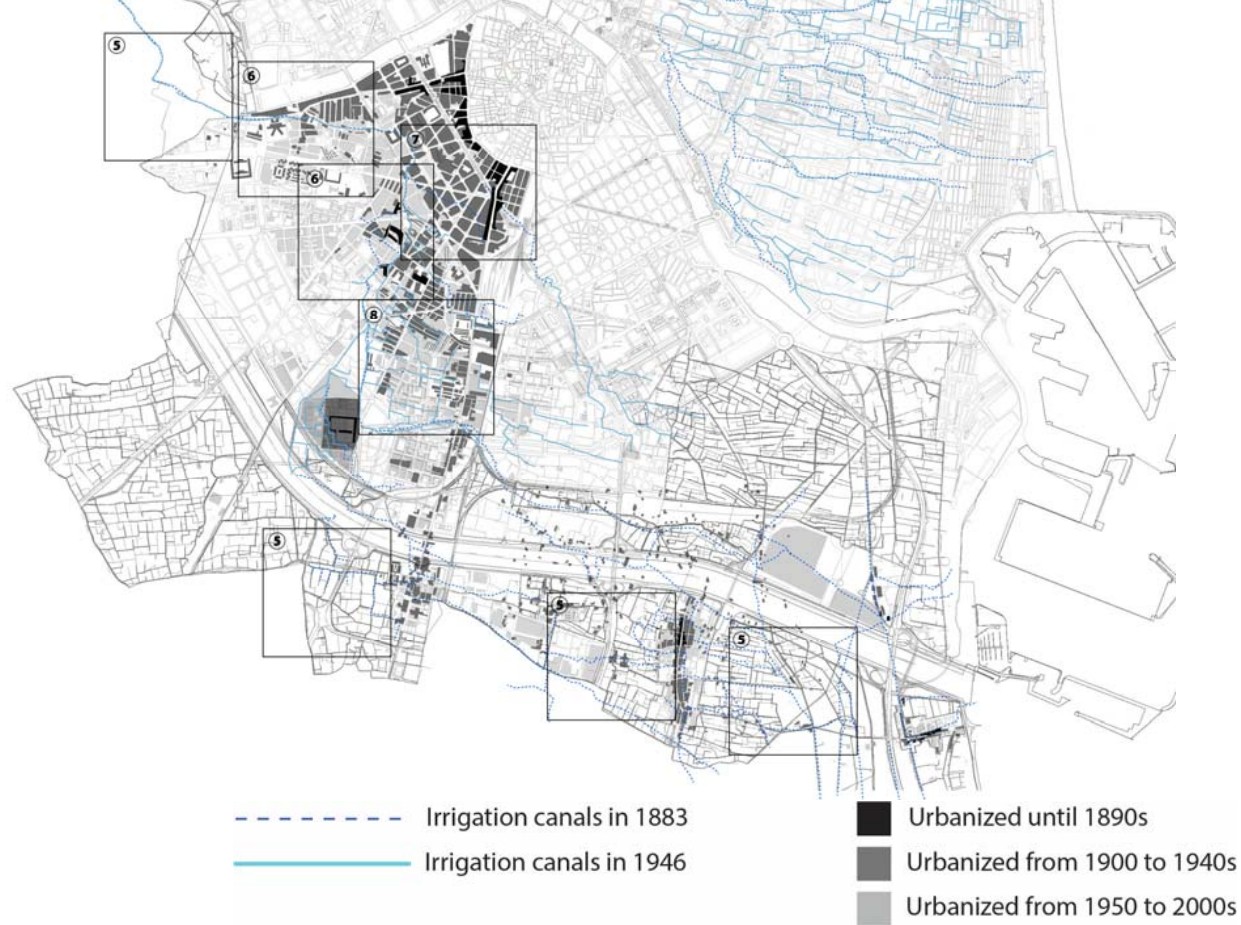

**Figure 9.** (⑤–⑧) The transformation of the irrigation canals and the urbanization phases, along the itinerary of Acequia de Favara.

Figure 10 shows the transformation of the irrigation canals in detail in the areas ⑤. These areas are located in agricultural areas, where the canals are still in use as an irrigation system. They are in turn defined as agricultural, ecological and environmental protection zones in the planning in force [37], so the rural parcel and fractal irrigation network has remained.

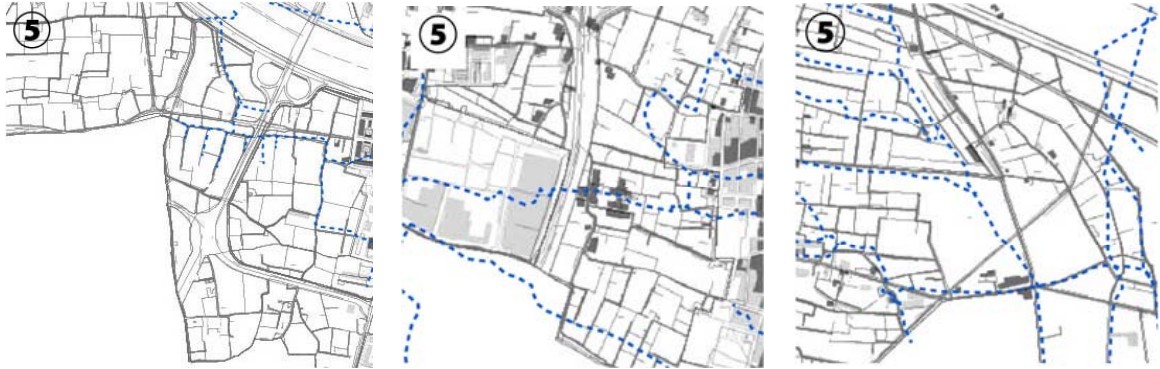

**Figure 10.** In the areas labeled as ⑤, the canals preserved from 1883 to 1946. These canals in these areas are still used for agricultural irrigation systems.

Figure 11 shows the transformation of the irrigation canals in detail in the areas labeled ⑥. In these areas, the historical canals remained by 1946. In some part, the irrigation canals were used as the border of urban expansion in the first half of the 20th century; therefore, they resulted in being kept untouched until 1946. Some parts of the canals cross the constructed area, and the buildings used the canal lines as their edge. These remnants suggest that the canals had priority over the functional form of urban space or the scale of urbanization for the society until the first half of the 20th century.

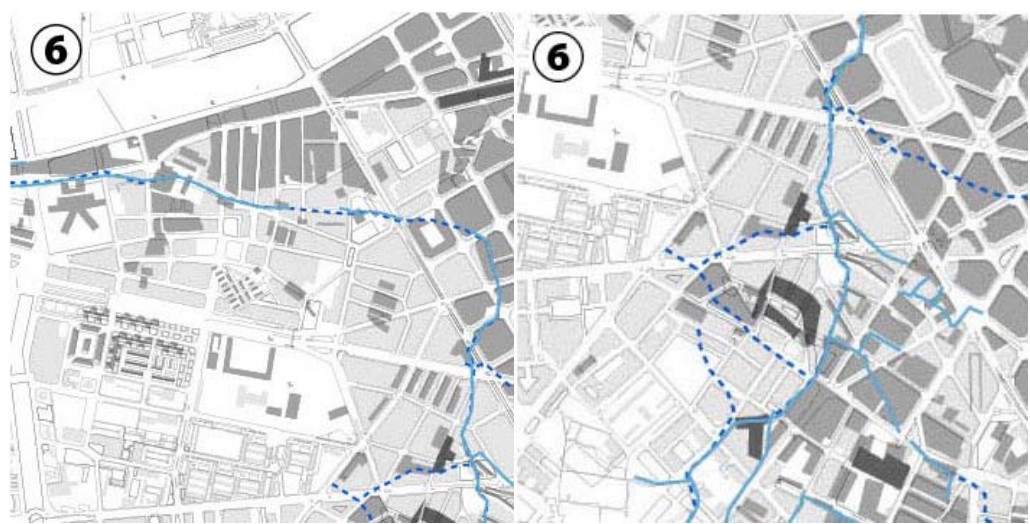

**Figure 11.** In the areas labeled ⑥, the canals preserved in urban space in 1946.

Figure 12 shows the transformation of the irrigation canals in detail in the area labeled ⑦. This area is one of those that developed before 1950 as residential areas. The canals that existed until 1883 were disappeared, and the functional straight lines were introduced as new building block patterns in the same area. It is estimated that the urbanization in this area was driven by rapid construction to meet the high demand for housing at that time, and the canals were considered to have less importance regarding the development of residential areas.

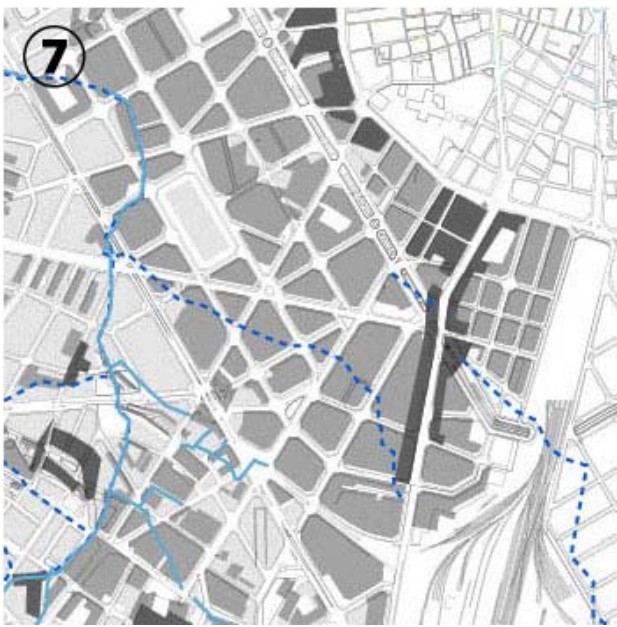

**Figure 12.** In the area labeled ⑦, the canals disappeared during the period between 1883 and 1946.

Figure 13 shows the transformation of the irrigation canals in detail in the areas labeled ⑧. In this area, the canals were expanded from 1883 to 1946, incorporated within the urbanized area in this period. The urban area of Valencia was enlarged to the direction of the South along with the highway and railroad in the 20th century; meanwhile, the construction of low-rise dwellings was increased gradually in this area. It is estimated that the speed and scale of these dwelling areas were not so high, and the settlers continued using the water supply from the traditional hydraulic systems. As a result, the historic irrigation canals were encompassed within the urban structure.

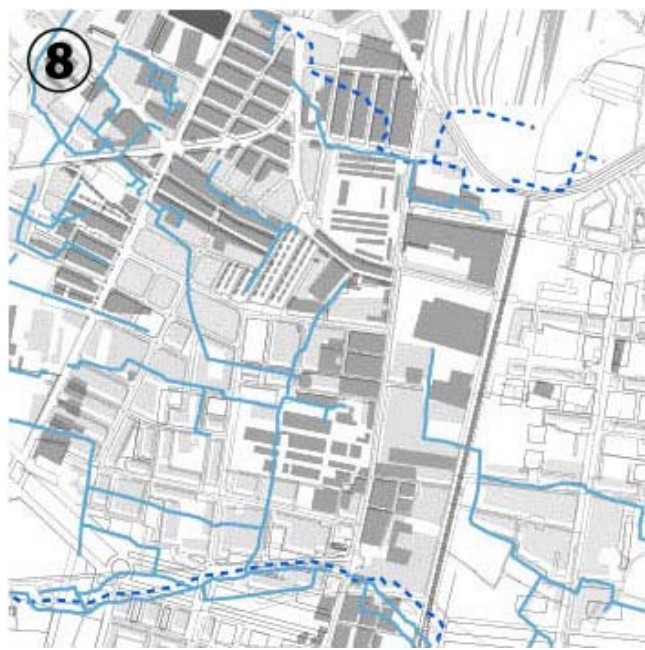

**Figure 13.** In the area labeled ⑧, the canals developed during the period between 1883 and 1946.

From a macro perspective, which we obtained through the study of documents on the two historical irrigation systems, a logic of centrifugal urban expansion is observed. Additionally, the differences in the process of urbanization depend on the time and place.

In the first half of the 20th century, the city of Valencia was expanded, integrating the historical peri-urban settlements. The historic suburban settlements literally inherited the canal routes in the urban morphology, and this peripheral expansion partially incorporated the paths and canals of the existing agricultural areas. Until the 1940s, in the regional scale, urbanization was a consolidated process of the development of the productive ecosystem and the societies of the settlements, which both were based on water usages, in spite of the homogeneous expansion of the city center. In other words, the continuity between the rural and urban landscape was maintained in urbanization in this period, due to the maintenance of the irrigation system between the historic settlements.

During urban development after the 1950s and up to the 2000s, less consideration was given to the canals. Much of the cultivated areas, which contained the anthropic watercourses, disappeared during urbanization in this period. Only the mother canals have maintained their shape and the water that they deliver, even though they are closed underground.

In addition to the above study focusing on the transformation of the canals, regarding the current urban morphology, we analyze the layout of irrigation canals and the linear features of urban configuration that appeared in building patterns and road networks. This exercise clarifies the difference of the relationship between irrigation canals and urban morphology in each modern urban expansion phase. Then, we examine the degree of the correspondence of the canal layout and the linear features in each phase of urban expansion before, during and after the modern urban expansion.

Figure 14, below, shows today's networks of the agricultural irrigation canals and the urbanized areas, segmenting it into the following periods: pre-1890s, 1900–1940s and 1950–2000s. In dark blue, we highlighted the irrigation ditch sections that correspond to the current urban morphology, for the canals of Favara and Mestalla. According to this identification mapping, we calculated the inheritance rate of the irrigation canals in the current urban structure, introducing, in, turn the temporal factor of urban development phases (Table 1). The index clearly shows that urban blocks created before the 1890s retained the shape of the canals to a greater extent than those created after the 1900s, and urban blocks created from the 1950 to the 2000s were less influenced by the canals.

*3.2. Micro Perspective Analysis: Deciphering the Urban Palimpset Based on the Present Urban Structure*

The field study was carried out with the aim of assessing the current state of the irrigation canals in the urban and peri-urban landscape, to introduce a research perspective from the micro scale. The macro perspective given by the document study allows us to examine the influence of the urban development on the transformation of irrigation canals. At the same time, the document survey shows that the urbanization process is a mechanism of accumulation/obliteration of the past. Thus, the present urban landscape is understood as an aspect of the constantly changing circumstance, as well as a scene of a point of time within the urbanization process, that is, the interrelation between inheritance and renewal of the past, including socio-economic, demographic, cultural, infrastructural and mobility layers [37]. Detailed observation of the present situation of the places where the historic irrigation canals were located is an approach to read the historical resources in the landscape and interpret the influence of irrigation canals on the present urban landscape.

**Table 1.** The inheritance rate of the irrigation canals in each phase of urbanization.

| | | Acequia de Mestalla | | Acequia de Favara | |
|---|---|---|---|---|---|
| | | Present Canals | Corresponding Lines | Present Canals | Corresponding Lines |
| 1950–2000s | Length (km) | 73.39 | 7.21 | 7.36 | 1.73 |
| | Rate (%) | | 10% | | 24% |
| 1900–1940s | Length (km) | 4.04 | 1.09 | 3.74 | 1.23 |
| | Rate (%) | | 27% | | 33% |
| –1890s | Length (km) | 1.88 | 0.64 | 1.52 | 1.34 |
| | Rate (%) | | 34% | | 88% |

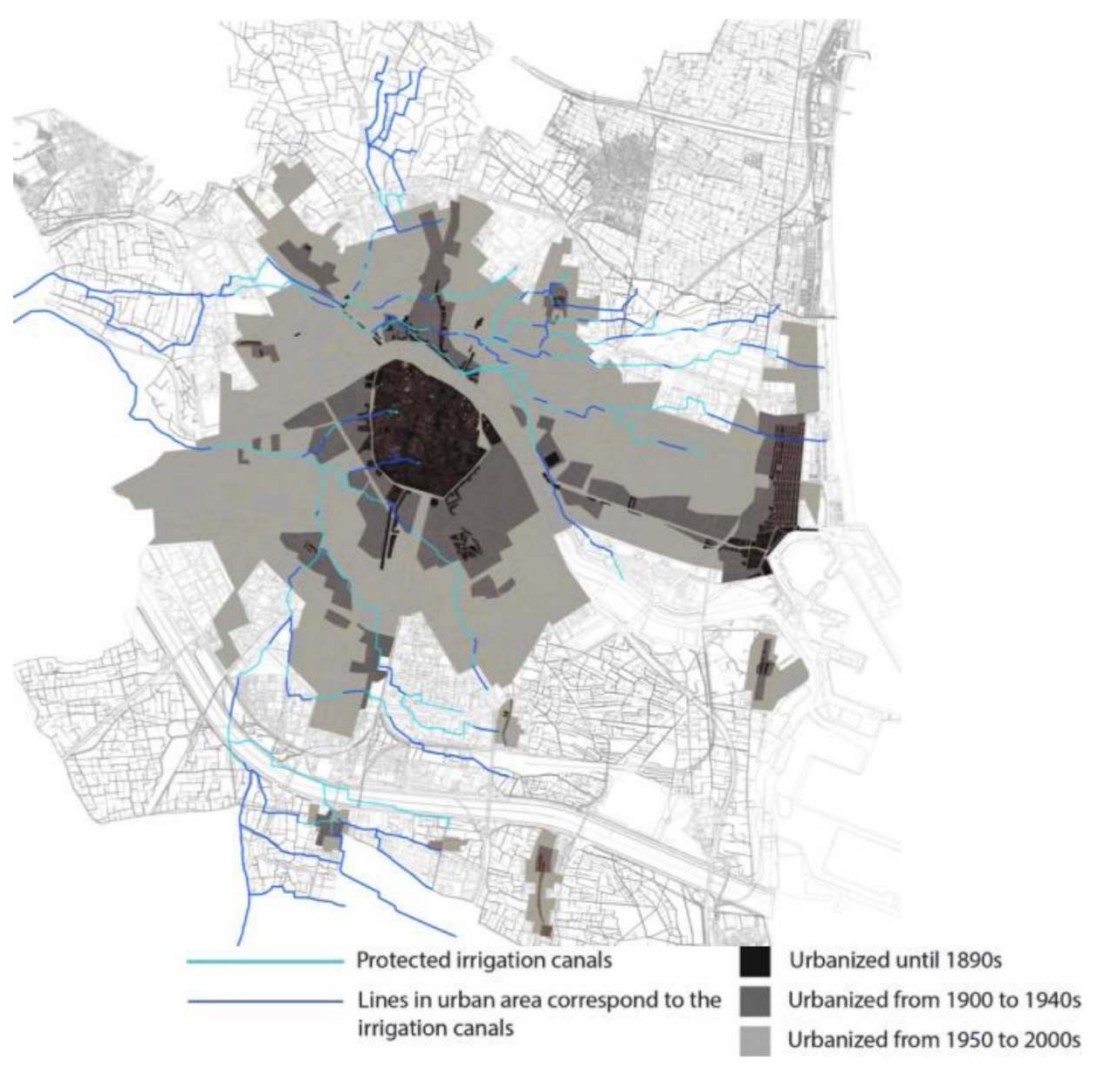

**Figure 14.** The inheritance of the irrigation canals and the phases of urbanization.

### 3.2.1. Acequia de Mestalla

In the territory of the city, the entire part of the target watercourse of the canal are reformed as tubes. Partially, the tubes are visible in the agricultural areas but in the urbanized areas, they are buried underground. Figure 15 shows the study targeted line of irrigation canal and the linear inheritance of tnem to the current urban structure.

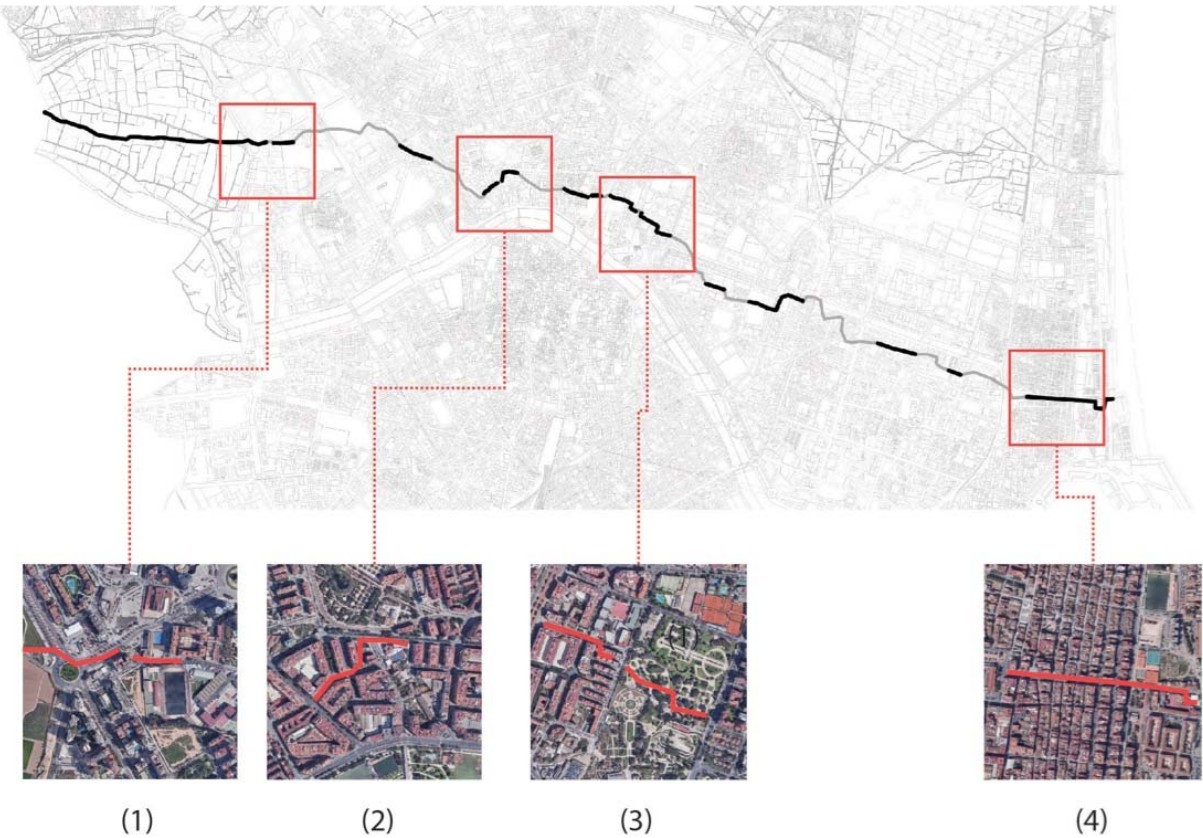

**Figure 15.** The inheritance of the irrigation canals in urban spatial structure and satellite images showing the type of land configuration in relationship with canal layout. (**1**) The inheritance located between urbanized area and cultivation area. (**2**) The inheritance located in urbanized area. (**3**) The inheritance located between building block and urban green. (**4**) The inheritance located in the historic settlement.

In the agricultural area, the main flow of the canals is invisible but their layouts are indicative in the crop patterns as ridges. There are tributaries around the crops with some sluices to take water from main flow. It shows that the water management system for the cultivation is still dependent on the manual operation by irrigation canals. Near the main flow, there are ancient bridge structures, weirs, and old factories. Moreover, some linear ground overgrown with reeds is observed, showing the abundance of ground water, which possibly have influence from the irrigation canals.

In the urbanized area, in some cases, one can read the transposition of the canal traces into the urban pattern. As a principle, the historical urban areas preserve this history of water in their morphogenesis because in the early ages, the city growth and the hydraulic system development were interdependent, both in their layout and in their irrigation logic. The modern or contemporary areas are more alien to the blocks designed in the form of a grid, which makes the hydraulic reading of these areas difficult. However, in the remarkable urban expansion and renewal, especially in the 20th century, the urbanized area retain partially the form of the canal lines in their town blocks, the shape of the streets or in the micro-topography. Some topographic characteristics of the canals, such as the gap created by reservoir, still can be observed in today's building patterns (Figure 16). Moreover, some urban green areas are found to be located in the flood prone area created by the ancient river bed of Turia, which previously was an important node of the irrigation system (Figure 17).

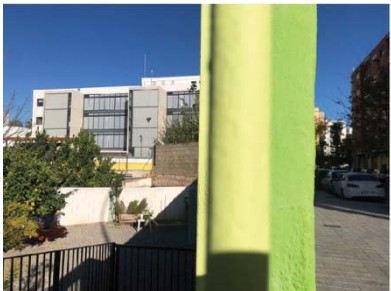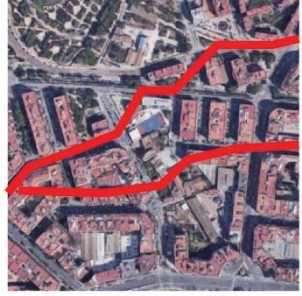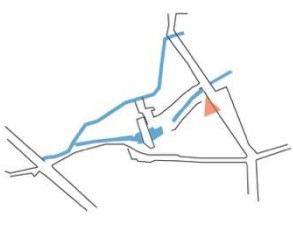

**Figure 16.** The image of the level differences in the urban area, the satellite image and the diagram of spatial structure created based on the cadaster created between 1929 and 1946 at the same location.

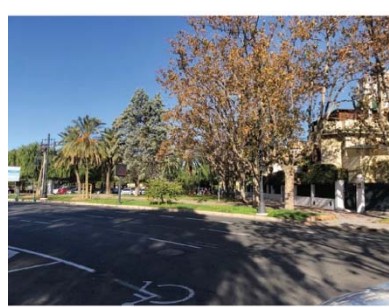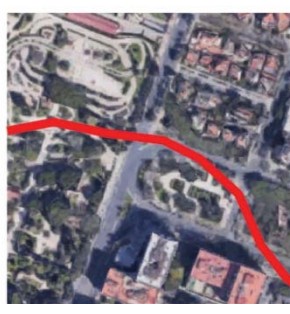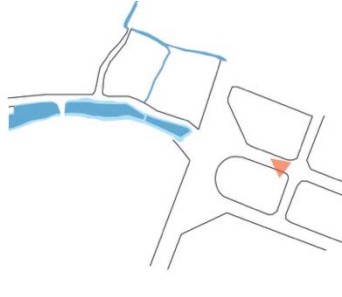

**Figure 17.** The image of a green area in the urban area, the satellite image and the diagram of spatial structure created based on the cadaster created between 1929 and 1946 at the same location.

### 3.2.2. Acequia de Favara

Figure 18 shows the study targeted line of irrigation canal and the linear inheritance of tnem to the current urban structure. In the urbanized area to the northwest of the city, even though the waterways are not present but exist as covered conduits, street lines that correspond to the line of the irrigation canals are observed. It is estimated that originally, the town blocks were constructed along with the historic canals first, then the canals were buried underground. The existing urban blocks constructed during the modern urbanization retained the original town blocks, and thus in this part, the line of the canal retained its shape in the urban structure as the base of urban planning, connecting the urban and green areas.

In the south of the city below the new Turia riverbed, the line of the canal is identical to the street lines, except for three discontinuations by the highways and the train line. Additionally, in the towns in this area, the town blocks kept the shape of the historic canals, continuing to the paths in the agriculture areas that correspond to the present canal line. That is, the water canal is closely related to the urban structure in these towns, keeping continuity between urban areas and agriculture areas.

To the west of the city, the waterways are not visible in the present urban structure, but there are urban green areas with the shape of the pre-existing canals, from which the pre-existent waterway can be traced. Furthermore, in the district of "Patraix", the church "Església del Sagrat Cor de Jesús de Patraix" is located on the Plaza of Patraix on the Convent de Jesús street (Figure 19). In addition, in the district of "Poblados del Sur", the "Parroquia Nuestra Señora de Gracia" church is located on Alba Street. These districts were created before the 20th century, together with these churches, and the town blocks were incorporated with the canals. In the modern urban renewal, these churches and surrounding building blocks and streets retained the original shapes; as a result, the shapes of the canals also remain in the present urban structure.

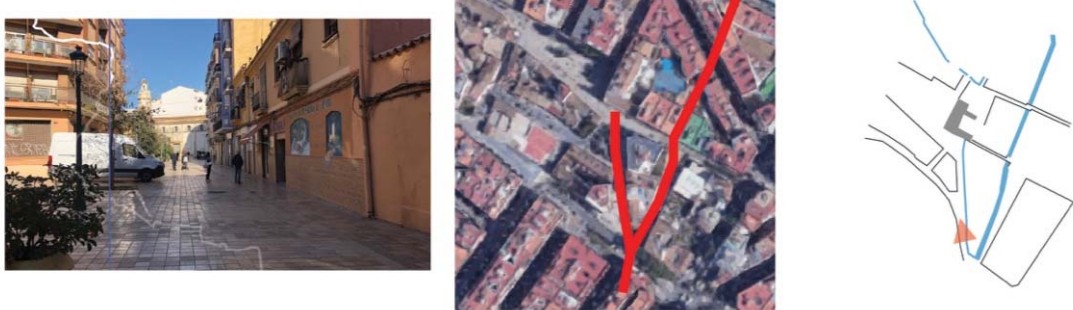

**Figure 18.** The inheritance of the layout of irrigation canals in urban spatial structure and satellite images showing the type of land configuration in relationship with canal layout. (**1**) The inheritance located between building block and urban green. (**2**) The inheritance located in the historic district and newly build riverbed. (**3**) The inheritance located in the industrial area. (**4**) The inheritance located in the historical settlement.

**Figure 19.** The image of a building block inherited the canal line, the satellite image and the diagram of spatial structure created based on the cadaster created between 1929 and 1946 at the same location.

Additionally, in the south of the city, the canal was partially refurbished as an open conduit and used as a landscape in the park "Parque de los jubilados". It is an example of the application of the historical waterway to urban design.

Moreover, some urban green areas are found to be located in the swale created by the micro topography of the city, where the densely reticulated irrigation canals existed before the urbanization (Figure 20).

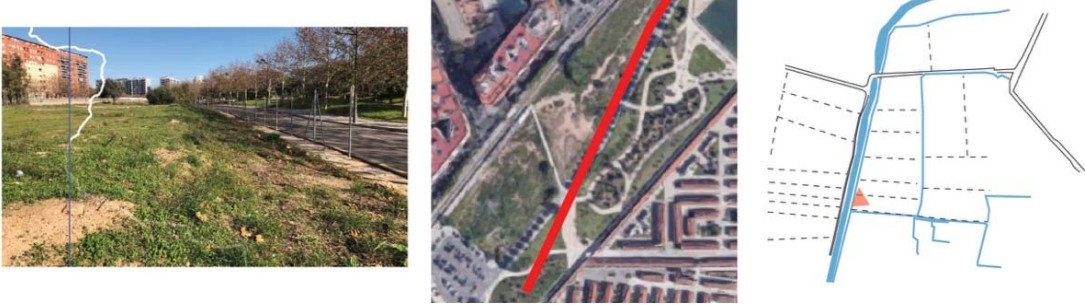

**Figure 20.** The image of a green space in the urban area, the satellite image and the diagram of spatial structure created based on the cadaster created between 1929 and 1946 at the same location.

### 3.2.3. Present Urban Landscape and Irrigation Canals

The field survey reveals the different manners of the influence of the irrigation canals on the present urban spatial structure. Figure 21 shows these typologies of the sections that illustrate the relationship between the irrigation canals and the present landscape. Group (a) explains the section patterns in urbanized areas, group (b) explains the section patterns in rural areas.

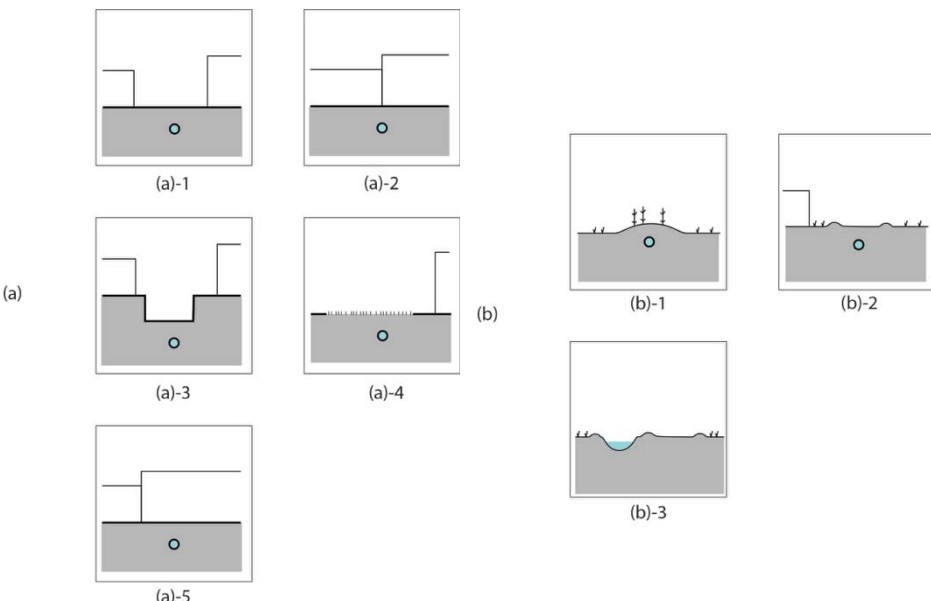

**Figure 21.** Typologies of the relationship between buried irrigation canal and current urban landform. Group (**a**) shows the section patterns in urbanized areas. (**a**)-1: The irrigation layout corresponds to the road layout. (**a**)-2: The irrigation layout corresponds to the plot inner separations. (**a**)-3: Sunken topography created with canals and current urban landform. (**a**)-4: Urban green transformed from floodplain of canals (**a**)-5: Current urban landform that is not related to the irrigation canal. Group (**b**) shows the section patterns in rural areas. (**b**)-1: The irrigation canal layout corresponds to the ridge of the earth between crops. (**b**)-2: The irrigation canal layout corresponds to the rural road layout. (**b**)-3: The irrigation canal that is currently used corresponds to the road layout.

In the urban areas, the direct influence from the irrigation canals is observed in the building layouts. In addition, there are indirect influences recognized on urban topography through the detailed perspective. Before the expansion of the modern city, the canal networks were incorporated into the town blocks. The historic canals were constructed to use only the natural gravitational force, thus the level differences, slopes or the sunken places are essential to ensure the certain water flow or to control and manage them. These geographical features were built in the city for water reservoirs or the diverging points together with the canals, using the original topography. In the urban development and renewal of the city in the latter half of the 20th century, the irrigation canals were buried, and the structure of the canals were filled up; the watercourse itself is not visible in the current urban landscape. However, these geographical features created by the amalgamation of original landform and anthropic technology remain, even they were overwritten by new technologies with new materials and buildings.

In the rural areas, the buried canal presence is perceived not only by the topography, but also by the presence of vegetation along the crops (reeds, arundo donax, pópulus alba, etc.). The canals constitute an element of hydration and infiltration along their course because the canals were drainage trenches dug into the land itself originally, thus forming a sort of anthropic river. Today, these trench canals still exist, although many sections have a quadrangular construction whose function is to minimize losses and ensure the continuous flow of water. Despite the fact that the water is channeled, the materials used are not watertight; therefore, infiltration is continuous along the length of the canal.

Moreover, the field study demonstrates that there are some aspects that show the imprint of the pre-existing canals on the urban landscape. Firstly, the existence of empty plots of land or plots of land where part of the irrigation ditch may have emerged, or where there is evidence of humidity of the soil, which encourages the growth of various spontaneous plant species in the manner of the "Tiers paysage" [38], is a first indication of interest. Furthermore, this detailed observation demonstrates that the irrigation systems create the topographical evidence in slope lines. By interpreting the spatial and topographical concatenation following the route of the canal, we can identify strategic slopes in certain streets, raised or depressed land, depending on their position in relation to the route of the historic canals.

These factors are, in our opinion, very important when we rethink the biotopical factor of urbanity, microtopographies, natural runoffs recovered and associated with the system of green spaces and the irrigation system, which are the issues that we intend to highlight through this research.

In a document study, we can affirm that urbanization, especially modern urbanization, has been damaging the permanence of these hydro-systems in the urban environment. However, in a field study, with more precise and subtle reading, it is possible to interpret urbanity from a hydraulic and biotopic perspective, and above all, we postulate that it is possible to reconnect urbanity to the landscape and the anthropic geography of irrigation canals.

## 4. Irrigation Canals and Urbanization

This study illustrates several threads of evidence, which highlight the relationship between historical irrigation canals and the spatial structure of urbanized areas, as well as the complete elimination of them in urban development and modernization.

It became clear that in the peripheral part of the historical settlements around the city center of Valencia, especially around the towns Benicalap, Patraix, El Cabanyal-Canyamerar, La Torre, Horno de Arcedo and Castellar-Oliverar, the town blocks correspond to the form of the irrigation canals. It shows that the urban development from the 1950s to the 2000s consist of the expansion of the existing settlements, as well as the creation of the new city area between the city center and its peripheral settlements. The peripheral part of the settlements was developed by extending the existing roads inside of the settlement and connecting them to the rural roads, which were closely related to the irrigation canals.

In this way, the newer town blocks, which share previously existing blocks, have also inherited their lines from the irrigation canals. The new areas between the city center and the peripheral settlements were created as the superscription of the grid pattern of town blocks centralized within the infrastructure, thereby erasing the winding path and the canals in rural areas [39].

Figure 22, below, shows the transformation of the irrigation canals around the neighborhood Patraix [33–35], as an example of the ancient settlement created and developed along with the irrigation canal in the 19th century, which was subsequently absorbed into the more recently developed area of the city. Patraix was a small village surrounded by cultivation areas in 1883, with the new city area around the existing settlement having been created according to the partial plan approved in 1950 by overwriting the grid pattern town blocks onto the rural areas.

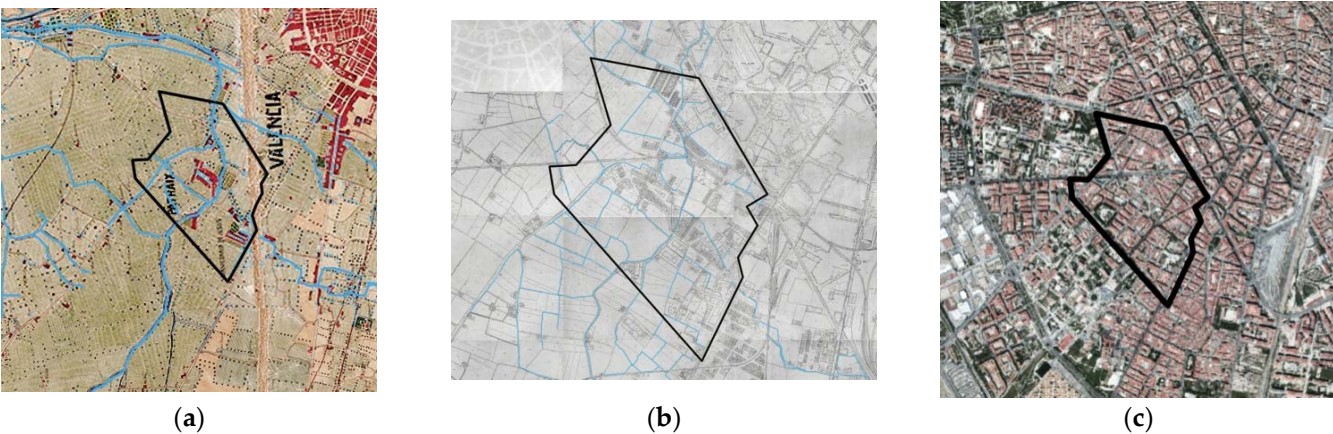

(**a**)　　　　　　　　　　　　　　　　(**b**)　　　　　　　　　　　　　　　　(**c**)

**Figure 22.** The neighborhood of Patraix within the cartography of 1883 (**a**), collected cadaster of 1929–1944 (**b**), satellite image of 2019 (**c**) [33–35].

Figure 23 shows the protected irrigation canals and pre-existing canals until 1946 in the present urbanized areas in Patraix. The original settlement was a small village, created based on the irrigation canals. Within its center, there are "plaza de Patraix" and a church. This neighborhood was completely urbanized, with the irrigation canals that existed until 1944 being buried and only two lines currently being protected. However, there are some lines of streets and town blocks that still correspond to the irrigation canals inside of the present urban structure. Figure 24 shows that the canals lines next to the building blocks that constructed before 1900 were inherited in the current urban configuration. On the contrary, the canals developed together with the expansion of the new building blocks until 1950 were demolished after the modern urbanization and the city was expanded further in a grid pattern.

Moreover, it shows the difference of urbanization by the location in this district. The urbanization in Patraix occurred both centrifugally and centripetally, with the area that was created by expanding the original settlement before and the section that was created as the new urban area in the latter half of the 20th century. The part created based on the original settlement before 1946 was developed along with the irrigation canals and, therefore, inheriting their lines. In contrast, in the area developed according to the grid blocks, the historical canals were disregarded.

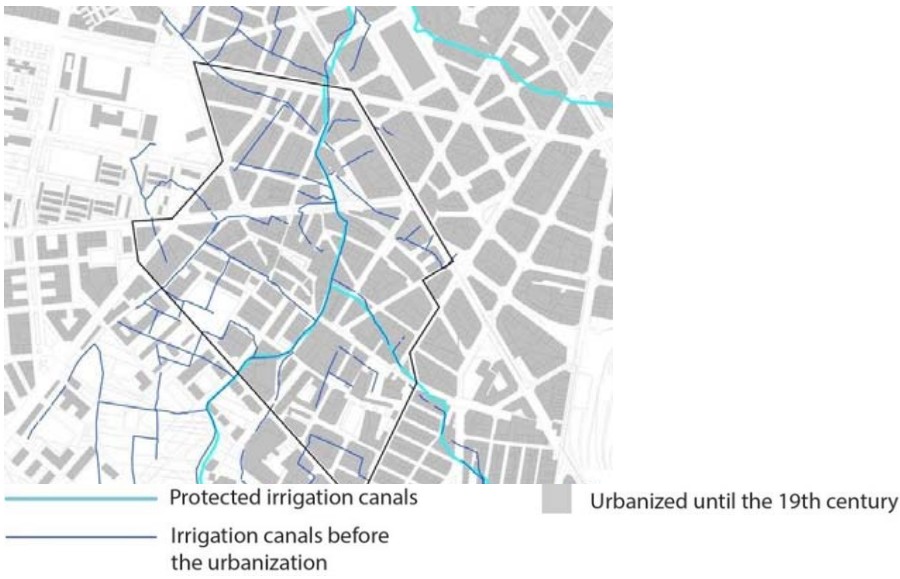

**Figure 23.** The transition of the irrigation canals in Patraix.

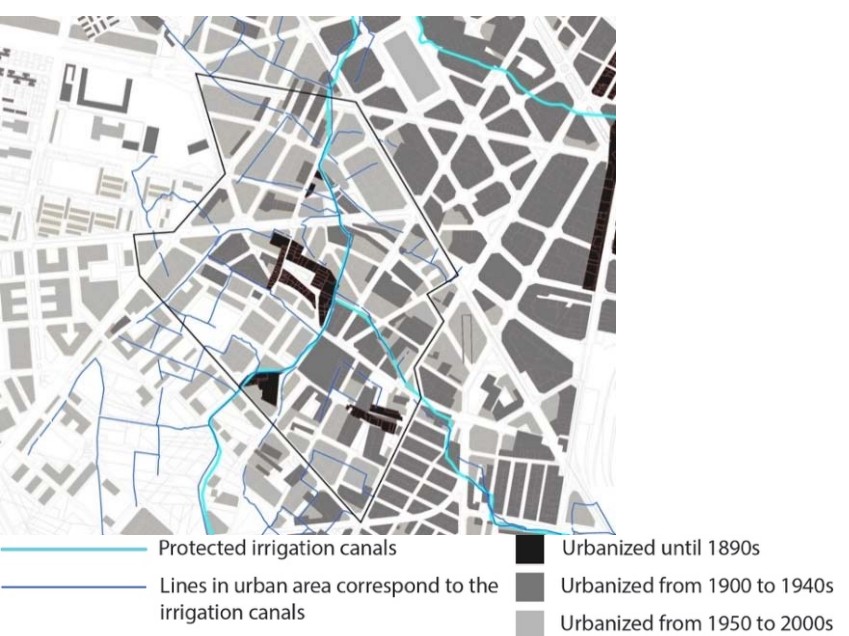

**Figure 24.** The transition of the irrigation canals and urban development in Patraix.

Furthermore, in the part that had dense irrigation canals until 1946, the urban plots are rather fragmental than in a grid pattern. And the urban greens are located on the areas correspond to the irrigation canal dense area or canal nodes (Figure 25). It proves that the irrigation canals and the urban spaces are indirectly related to each other. The details regarding the remaining lines of the canals, the differences in the form of their remains, their influences on the urbanization process and whether there are differences depending on specific areas and so on, are left as possible objectives of future research.

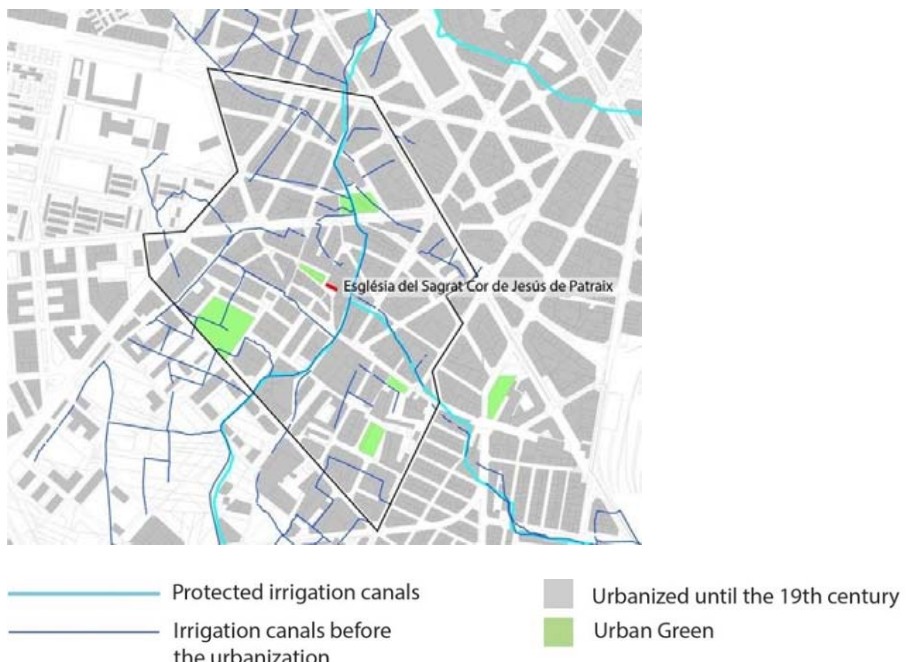

**Figure 25.** The transition of the irrigation canals and urban greens in Patraix.

## 5. Discussion

On the basis of the aforementioned studies, the connection between historical irrigation systems and urban morphology were established within the scope of the urban spatial structure, urban development process, and social structure.

Concerning the urban structure, the historical irrigation systems are hidden underground in the urban areas and separated from today's urban context, functionally and visually, yet they are partially transcribed into the urban landscape by means of alignments, maximum slope lines, or wet soil systems. It is because the hydraulic layouts, based on the gravity-flow and environment-friendly technology, interpret and modulate the geographical conditions of the territory.

Concerning the urban development and its process, the degree of influence of the historical hydraulic systems on urban morphology is inversely related to the scale and speed of urban transformation. The areas that experienced rapid urban development with a large construction scale do not contain directly the organic shapes inherited from the historic canals. On the contrary, the areas that grew gradually with a little amount and scale of construction maintain the historic shapes and, sometimes, the visible water courses.

Regarding the social structure, the irrigation canal lines retained their shapes in the forms of several historic buildings that are currently occupied and in use. The influence of the canals on the creation of social and cultural space is assumed. Moreover, the inheritance of the historic water infrastructure to the underground water connection may have affected the land and building uses in the city, such as parks, public institutes, historical markets, pools or tannery factories, etc.

The present study and the studies completed by the co-authors bring the reevaluation of the canals in the context not only of the rural landscape, but also of the present urban landscape. The irrigation canals are not merely the past heritage of Valencia that founded the fertile land and were dispelled by urban development to meet the housing demand and economic growth. Regarding the historic center, the hydraulic systems and the geology and topography shaped by them affected the urban vegetation and spatial characteristics [5]. In addition, the canals influenced the creation of the new town blocks in the expansion of the city center directly and indirectly in the 19th century by their shape [6]. The present study highlights the geographical relationship between the canals and urban spatial structure in the further urban expansion, and the influence of canal systems on the present urban spaces.

Yet, the detailed examination of the transformation of irrigation canals from various aspects, such as economic, political, urban health and safety, living environment and activities of the local communities, is essential to gain an intrinsic understanding of the relationship between canals and the city, or the water and the inhabitants.

Cities were created on the basis of water, both for it and against it, i.e., by protecting themselves with all kinds of hydraulic infrastructures, such as canals, dikes, diversions, channeling, etc. The existing cases of urban river restoration in the world, such as Albufeira in Portugal [40] or the restoration of the Cheonggyecheon River [41] in South Korea, allow us to hope that such initiatives of recovery or partial reversion, with the aim of introducing ecosystemic variables into the environment, are possible. The case of Banyoles [42] in Spain is a paradigmatic case of the revaluation and reopening of the network of irrigation canals in the historic urban environment.

Regarding the rehabilitation of indigenous ecosystems and urban development, there is a social housing project of Ciudad Verde in Bogota, the capital of Colombia. This project is managed by the private sector, conceived as a self-sufficient town with a mixture of uses for an expected population of 200,000 inhabitants [43]. Since the project is located in a flood prone area over the Bogota River Basin, water management included mitigation measures that are mainly the heightening of existing dikes along agricultural canals, and the construction of a separated storm water drainage with a series of retention basins and pump stations [43]. Originally, this watershed was reclaimed by the pre-Hispanic settlement as an extensive agricultural area. It was based on the creation of the ridged fields and irrigation canals using the indigenous micro-topography there, before the colonization. However, urbanization by colonization created a huge gap between the society of the urban areas of the foothills and the rural areas of the watershed. Moreover, the concrete canal construction after 2000 hindered a healthy perception of and relationship with water [44]. The main problem of the new canal is the faulty connections between the sewage and the storm water system. As a result, untreated direct discharges are the major source of pollution, causing urban fragments as well as social issues, insecurity and detachment to the water infrastructure.

This is an on-going project; its impact on the society and natural environment is still not clear, yet it might be an interesting example of sustainable development toward the rehabilitation of the ecosystem and mitigation of environmental threats based on the water management and historic irrigation canals.

Apart from the direct reusing or rehabilitation of the irrigation canals, other studies on their influence of topography on urban morphogenesis [45] imply the possible suggestions of the usage of buried canals with biotopical and circular parameters. The khettars or qanats of Iran [46] or Morocco [47] serve as punctual vents, contributing to form a circular system [48], using soil and infiltration as part of it.

Back to the case of Valencia, regarding the future usage of these historic canals for sustainable urban development of Valencia, an interesting course of action on these canals could be, first of all, a definitive separation from the sewage network, in order to preserve their cleanliness as much as possible. Secondly, once cleaned (which is not so easy to achieve) the network of canals could be the basis for the reconfiguration or implementation of the urban green infrastructure network. In this practice, it is indispensable to use the topographical, geological and biological characteristics of the rural landscape created by hydraulic systems. The combination of these characteristics with the canal related structures and constructions, such as old weirs, bridges or rural houses, may have potential for the implementation of sustainable urban development. The incorporation of historic canals and present urban spaces encourage the maintenance of green/blue spatial continuity and coherence in the landscape, while proactively integrating or rooting the historic and heritage elements of the city's water memory. Figure 26 shows the urban green, vacant plots, micro topography together with the previous canal lines of Mestalla and Favara. This map shows the potential of recovering the green and blue infrastructure in the city, connecting them each other including our society and creating the new sustainable ecosystem.

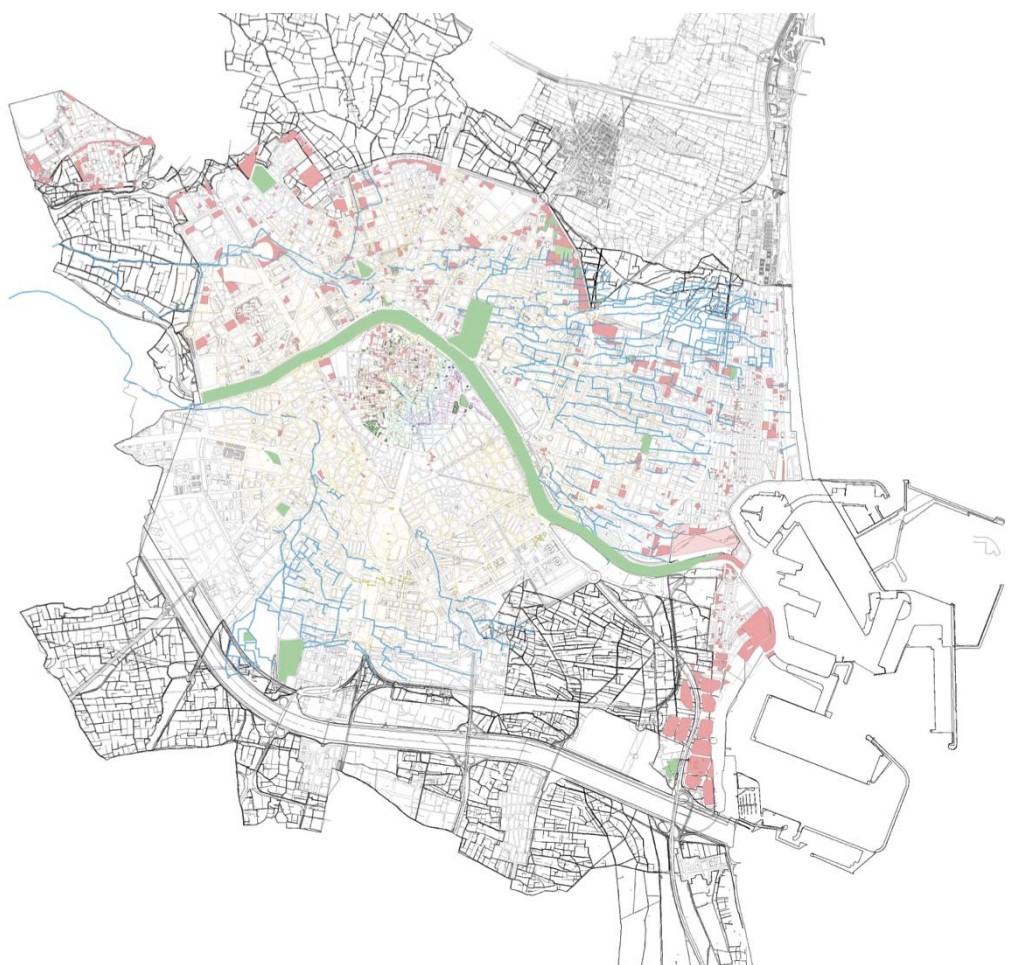

**Figure 26.** The city with the present contour lines, unoccupied plots and green areas with lines of irrigation canals from the cadaster 1946.

Finally, this incorporative perspective of urbanity implies an understanding of the city as a human habitat inserted into an environment, in a territory, and not isolated or artificialized. In this sense, the research proposes to delve into the potential alliances between the biophysical matrix and urbanity. This perspective is in line with the increasingly evident considerations and points of view concerning environmental protection, sustainable urban development and urban and environmental ecology. Furthermore, the establishment of water separation systems with respect to urban runoff, irrigation water, as well as rainwater, is technically imperative.

## 6. Conclusions

This research explores the relationship between urban morphology and historical hydraulic systems by analyzing the evolution of historical irrigation canals and urban development in the 20th century through document survey and field survey. Additionally, our study suggests, for the mechanism of Valencia's urbanization, the possibility that it consisted of the interrelationships of three movements: the expansion of historical settlements, the expansion of the city center, and the sprawling of the city territory by creating the new areas.

In the first half of the 20th century, the historic center was expanded rapidly with a homogeneous grid pattern, and peripheral settlements were also expanded, though it was by the gradual extension of the existing areas along with the rural plots. As a result, the irrigation canals were inherited in the peripheral areas of the historic settlements, related to the speed and scale of the urban developments.

In the second half of the 20th century, the city was expanded largely by the creation of new territory with functional patterns. As a result, the canals were mostly obliterated in the urbanized areas, and they are not visually indicative in the urban landscape. Nevertheless, the detailed examination by field survey shows that the topographic or the geological characteristics of the canals are inherited in the present urban space.

Consequently, this study illustrates the connection between the current spatial structure of urban areas and the previous rural landscape, as well as the interaction between today's urban landscape and rural landscape, by means of historic irrigation canals.

Furthermore, several possibilities for future research are suggested, clarifying not only morphological issues, but also questions concerning the evolution of society, urban ecology and the biotopical character of urbanity. Questions of an urban nature or urbanity have important relevance.

It is generally accepted that shared spaces in the city, over time, provide the possibility for a healthy and flourishing society. Prior to modernization, the canals of Valencia's historic hydraulic system were strongly related to the city's spatial structure and the social spaces of its urban areas, together with its most vital element, water, as they connect its history with the present city and its community. However, the city of Valencia has erased its traces by separating itself from its biophysical matrix in the modern urban development. The present study corroborates this distancing, while at the same time, in equal measure, it contributes to finding fragments of reversion, or possible reparation, in a search for desirable horizons to root our contexts of life in the territory. Additionally, the researchers and the new initiatives are trying to review the limits or reevaluate the interactions between the contemporary city and the memory of water associated with a social structure that roots us in the territory and contributes to the theory of urbanism, yet these actions are not integrated into the administrational or governance structure. It is a question of assessing whether these retro-prospective hypotheses focusing on the rural rooted systems supported by new technologies can be reintroduced with economical values, orienting ourselves toward the notions of landscape urbanism, and biotopic urbanity. It is hoped that future analyses and examinations, based on the hypotheses that this article has examined to clarify the unique urbanization mechanisms of the city of Valencia, with respect to future urban design, will set an example that will contribute to sustainable urban development and urban renewal.

**Author Contributions:** Conceptualization, K.S. and A.T.A.; methodology, F.I. and K.S.; formal analysis, F.I. and K.S.; investigation, F.I.; resources, F.I., K.S. and A.T.A.; data curation, A.T.A.; writing-original draft preparation, F.I.; writing—review and editing, F.I., K.S. and A.T.A.; visualization, F.I.; supervision, K.S. and A.T.A.; project administration, F.I.; funding acquisition, K.S. and A.T.A. All authors have read and agreed to the published version of the manuscript.

**Funding:** This research was funded by Japan Society for the Promotion of Science, grant number JP18H05449.

**Institutional Review Board Statement:** Not applicable.

**Informed Consent Statement:** Not applicable.

**Acknowledgments:** This work was supported by the French State through the National Research Agency under the "Investissements d'avenir" programme with the reference ANR-17-CONV-0004. We thank Jason Winther for English proofreading.

**Conflicts of Interest:** The authors declare no conflict of interest.

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
