# Peer review of "The Influence of Historical Irrigation Canals on Urban Morphology in Valencia, Spain"

_land, doi:10.3390/land10070738_

Round 1
Reviewer 1 Report
The authors are to be complimented with the improvements made in terms of balance and clarity of presentation.
Yet, still a significant number of (minor) language issues such as misspellings and plural/singular mismatches remain, see the list below.
Also, only with close viewing after significant expansion on screen, I was able to decipher most of the details, but I wonder whether the level of detail of several of the cartographic Figures (for example, but not only Figure 21) will work out in a printed/paper version; the authors may think of another solution for a paper version.
Overall, the author’s attempts at generalization, resp. comparing the findings and historic patterns for Valencia to other cities or urban regions, remain limited. As a result, the paper’s focus (and author’s intention) remains on a detailed analysis and data presentation for Valencia only, making it interesting for a relatively small number of readers only. For example, a discussion on the generalizability of the five hypotheses presented at the beginning of section 5, even if speculative, would be most welcome. It should not be too difficult to find indications about their validity for other cities/urban regions, even without doing the very detailed analysis as presented here for Valencia.
I further offer the following detailed suggestions:
As a service to the authors, please note spelling and/or grammar issues in lines 42, 48, 49, 100, 110, 163, 233, 344/5, 394, 402, 448, 450, 456, 484, 519.
Figure 6: suggest to enhance contrast
Figure 29: make boundary of research area more pronounced and visible in the top, right map which is already difficult to compare given the scale difference with the other the other three maps in this figure
Author Response
Thank you very much for your comment.
Please see the attachment.

Reviewer 2 Report
This is a well-developed and presented work that deserves to be published. Nevertheless, a throughout spelling check is necessary. The text includes too many typos. Please, have a close look!
For example:
42 – orderrther
391 – Tthis
399 – Mmuch
… and many others!
In the methodological part, it would be interesting to include more information on how all documents and maps have been analyzed/reworked. For example, what kind of software has been used? Why?
A final self-reflection on the limits of this study and its international relevance (or not) should be added.
In regards to this statement - "Regarding the future usage of these historic canals for sustainable urban development, an interesting course of action on these canals could be to uncover the closed canals and incorporate them as urban design elements, so that the historic canals are integrated into the current urban green infrastructure, connecting past, present and future" -> here a more critical approach is necessary to your statement ways more sound. Is it really so easy to uncover the closed canals in urbanized areas? What kind of problems/limits does this action imply? Are there some successful examples of actions of this kind in Valencia or elsewhere that readers can take into consideration?

Author Response

(The authors gave the same response as above.)

Reviewer 3 Report
The study highlights the relationship between the historical hydraulic systems and the more recent urban spatial structure, with the focus on Valencia, Spain. The aim is pertinent. The urbanization of many cities influenced historic hydraulic systems and natural streamlines. In contrast, other cities are developed without considering the obsolete hydraulic infrastructure, and streams were canalized and transformed into culvert pipes. A research hypothesis was not clearly stated in the text of the introduction. However, it is evident that the authors considered as hypothesis that “Historical Irrigation Canals influence Urban Morphology in Valencia”.
This research is rich in information. It was presented a lot of maps and historical documentation. Meanwhile, most of the knowledge about the hydraulic systems was from modern times and not from the Islamic period that is supposed those systems were introduced in this area. Also, it is missing information about the pre-Islamic period. The history of agriculture in Europe does not start from the Islamic period and Valencia should have hydraulic infrastructure before that period.
The chapter state of the art about vernacular hydraulic systems is insufficient. It is based mainly on local documents and barely mentions international research and similar case studies outside the Mediterranean.
The methodology is based on document surveys and field surveys. The document survey methodology does not have a clear structure and does not follow a clear chronologic order. A high number of maps and aerial photos was presented without being clear that the authors have permission to publish this information that is not of their ownership. The above can create copyright issues. It is the authors' responsibility to have written permission to publish those documents.
Figure 6 does not show any clear information and could be deleted.
The objective of the field study does not show a clear connection to the aim of the research. For example, in line 296 it is mentioned “the continuity of these buried hydraulic flows can constitute a tool for new urban design.” Meanwhile, the aim was to verify if those hydraulic systems influenced the urban morphology of Valencia in the 20th century and not to create new urban design.
The results did not present clear evidence of influence in urban morphology. The aerial photos and maps show that most of the new urban areas follow straight lines. In contrast, the traditional hydraulic systems followed the morphology of the landscape. Figure 10 shows the transformation of the irrigation canals to urban área practically without taking into consideration the hydraulic heritage. Figures 11-14 confirms the above with preservation of some structures when they are used for agriculture purpose, abandonment when obsolete and development of new channels when there is need for agriculture in the peri-urban area.
From the results presented in this study is not clear if there are and where they are those vernacular hydraulic systems.
Table 1 is not clear what is presenting. Some of those channes are more recent. What is the length of the Islamic period channels? How much of this remain?
In Figures 21, 25 is not visible what was the irrigation iheritance ifluence, maybe it is needed stronger colors or thicker lines.
Figures 23, 24 and 27, 28 does not show a systematic influence of the irrigation system to the development of the city because sometimes is a green area, sometimes a road and otherwise has a building.
In discussion are presented the five hypotheses, but this should be presented in the introduction or methodology.
Those five hypotheses present the subjective conclusions of the authors and are not discussed in comparison to other studies. They are subjective because it was not followed a systematic methodology to answer each one of them. For example, the 5th hypothesis: how many calas are refurbished and incorporated into the landscape design of the open spaces? Maybe 1% of the open spaces, maybe 50%, but it was not presented in a systematic research.
Line 594 is an incomplete phrase.
In line 596, Regarding the historic center, the irrigation systems influenced the urban vegetation and spatial characteristics? I don’t know from where come to this statement because Valencia is known to be from the cities with less urban green spaces.
In general, Valencia is not a good example of city developing an urban green infrastructure that will protect it from natural disasters. Also is not a good example of respecting the natural heritage as they choose to blame the river for the floods caused by impermeabilization and deforestation. The city chooses to change the river course to an artificial channel at the south of the city instead of using the irrigation system of the floodplain. The city could develop with more green infrastructure that will decrease the runoff and risk of flood and could use the old irrigation channels and agricultural land as retention basin, but the urbanists of the city took into consideration neither the cultural nor the natural heritage of the urban area. Thus, in the conclusions chapter, I am afraid I have to disagree that the "Valencia Model" helps us contribute to the theory of urbanism. Maybe only as a bad example of rapid urban growth disrespectful to natural and cultural heritage. It is not by accident that UNESCO considers as immaterial World heritage the institution Valencian Water Court.
Meanwhile, I strongly agree with the last phrase of the authors in this paper: “these systems, supported by new technologies, can be reintroduced, orienting ourselves towards the notion of landscape urbanism.”
I think this last line represents the focus of this research about how to correct the mistakes of the 20th-century urbanization in Meditteranean cities and in the case of Valencia, how to restore its most vital element, water, as they connect its history with the present life in the city.
Many cities try to correct the 20th-century urbanization mistakes by restoring the river and the other hydraulic systems canalized and transformed in culvert pipes. One example is the Cheonggyecheon river restoration project and others that can be read in Land 2018, 7(4), 141; https://doi.org/10.3390/land7040141, Mehdi F. Harandi, Marc J. de Vries; An appraisal of the qualifying role of hydraulic heritage systems: a case study of Qanats in central Iran. Water Supply, 2014, 14, 1124–1132. doi: https://doi.org/10.2166/ws.2014.074, Ricart, S., Ribas, A., Pavón, D., Gabarda-Mallorquí, A. and Roset, D. Promoting historical irrigation canals as natural and cultural heritage in mass-tourism destinations. Journal of Cultural Heritage Management and Sustainable Development, 2019, 9, 520-536. https://doi.org/10.1108/JCHMSD-12-2017-0089, J. Hermosilla Pla, S. Mayordomo Maya; A methodological system for hydraulic heritage assessment: a management tool. Water Supply 1 May 2017, 17, 879–888. doi: https://doi.org/10.2166/ws.2016.186 or in papers about the Ancient Irrigation Systems of the Aral Sea.
I do not recommend accepting the paper in the current state because it is misleading, but I will recommend resubmitting the paper after improving the focus of the research to landscape urbanism, with fewer maps and more focused photos from the field study. Also, photos showing evidence of landscape urbanism. Also, the state of the art should be improved with more international references. Finally, it is needed to present more quantified results as in Table 1 and better discussion.
Author Response

(The authors gave the same response as above.)

Reviewer 4 Report
Dear editor,
Thanks for the invitation to review this interesting manuscript.
In my opinion, this work is well conducted and presents solid scientific soundness; besides, this article could provide enrichment to the thematic field.
Nevertheless, the authors should add references to similar studies and research to the discussion section to foster the debate over the topic.
Regards,
Author Response
Thank you very much for your comment.
Point1: the authors should add references to similar studies and research to the discussion section to foster the debate over the topic.
Regarding the comparison to other studies, focusing on to the relationship between irrigation canals and urban morphology, we added the case study in line 125-133. Also, in the discussion part, we added the examples of irrigation canal inheritance of the current urban morphology and their sustainable way of rehabilitation/reuse in line 652-681. In addition, we included the consideration to the application of the method of rehabilitation/reuse to the case of Valencia.

Round 2
Reviewer 3 Report
The authors have updated point by point the text as required by the reviewers. The modifications are correct and comprehensive. In addition, they have introduced new data and references to foster their conclusions. The authors have made a remarkable effort to improve the quality of the article, which can be accepted as it is.